# Genome-Based Mexican Diet Bioactives Target Molecular Pathways in HBV, HCV, and MASLD: A Bioinformatic Approach for Liver Disease Prevention

**DOI:** 10.3390/ijms26188977

**Published:** 2025-09-15

**Authors:** Leonardo Leal-Mercado, Arturo Panduro, Alexis José-Abrego, Sonia Roman

**Affiliations:** 1Department of Genomic Medicine in Hepatology, Civil Hospital of Guadalajara, “Fray Antonio Alcalde”, Hospital 278, El Retiro, Guadalajara 44280, Jalisco, Mexico; leonardo.leal3084@alumnos.udg.mx (L.L.-M.); apanduro53@gmail.com (A.P.); alexisjoseabiology@gmail.com (A.J.-A.); 2Centro Universitario de Ciencias de la Salud, Universidad de Guadalajara, Sierra Mojada 950, Col. Independencia, Guadalajara 44340, Jalisco, Mexico; 3Programa Doctoral de Biología Molecular en Medicina, Centro Universitario de Ciencias de la Salud, Universidad de Guadalajara, Sierra Mojada #950, Col. Independencia, Guadalajara 44340, Jalisco, Mexico

**Keywords:** liver diseases, hepatitis B, hepatitis C, MASLD, MASH, HCC, chronic disease, metabolic syndrome, Mexican diet, nutrigenomics, bioinformatics, enrichment analysis

## Abstract

Viral hepatitis B and C (HBV and HCV) and metabolic dysfunction-associated steatotic liver disease (MASLD) are major public health concerns in Mexico, driving liver cirrhosis and hepatocellular carcinoma. The Genome-based Mexican (GENOMEX) diet, rich in bioactive compounds, may provide a nutritional strategy for preventing and managing liver disease. This study combines a literature review with integrative bioinformatic analyses to map the antiviral and hepatoprotective mechanisms activated by GENOMEX-derived bioactives and assess their therapeutic potential for preventing and managing liver disease. A literature-based review integrated with bioinformatics to identify the pathways activated by nutrients and bioactive compounds of the GENOMEX diet against HBV, HCV, and MASLD, incorporating data from in silico, in vitro, in vivo, and clinical studies, was conducted. An integrative bioinformatic approach, incorporating the Comparative Toxicogenomic Database and Functional Enrichment Analysis (STRING, DAVID, and Enrichr), was used to identify links between genes, nutrients, and bioactive compounds, with a subset of Mexican food staples included in the GENOMEX diet. The GENOMEX diet includes bioactive nutrients that may modulate molecular pathways related to immune response, oxidative stress, nutrient metabolism, and inflammation. Through integrative analysis, we identified key molecular targets—including TNF, PPARA, TP53, and IL6—that are implicated in viral replication, MASLD progression, and hepatocarcinogenesis. Functional enrichment revealed that these traditional Mexican foods and their nutrients are associated with genes and pathways involved in viral infection, metabolic dysfunction, fibrosis, and liver cancer. These findings highlight that the gene–nutrient interactions of the Mexican staple food in the GENOMEX diet can be integrated into nutritional strategies to prevent and manage HBV, HCV, and MASLD, while reducing fibrosis and HCC progression. These strategies are especially relevant in regions where antiviral treatments are limited due to high costs, antiviral resistance, and an escalating mismatch between the population’s evolutionary genetics and modern environment.

## 1. Introduction

Chronic viral hepatitis and metabolic diseases are significant global health concerns, each affecting hundreds of millions of individuals worldwide. Firstly, as of 2022, approximately 304 million people globally live with chronic hepatitis B (HBV) or hepatitis C (HCV) infection. These viral infections mutually account for over one million deaths annually, mainly due to cirrhosis and hepatocellular carcinoma (HCC) [1,2,3]. Secondly, the global prevalence of metabolic syndrome and its associated comorbidities, including hypertension, cardiovascular and renal diseases, type 2 diabetes, metabolic dysfunction-associated steatohepatitis (MASH), and the inflammatory progression of metabolic dysfunction-associated steatotic liver disease (MASLD), has steadily risen over the last decades [4]. It has been documented that metabolic syndrome is present in approximately 25% of the population in high-income countries, with prevalence increasing with age [5].

Furthermore, as part of the ongoing epidemiological transition in Mexico—from a predominance of communicable diseases to an increasing burden of non-communicable chronic diseases—it has become increasingly common to encounter patients with overlapping viral hepatitis and MASLD [6]. In both cases, pathways linking infectious and metabolic chronic inflammation activate cytokines, induce oxidative stress, lead to mitochondrial dysfunction, and promote dysbiosis. Furthermore, both conditions weaken the immune system due to persistent immune responses that trigger liver fibrosis/cirrhosis, which may ultimately lead to HCC [7]. These conditions pose additional challenges in managing liver health that require attention, as metabolic conditions can exacerbate liver disease progression in hepatitis patients [8,9,10].

In Mexico, the impact of these diseases—whether occurring alone or in combination—is magnified by restricted access to antiviral therapies, high healthcare costs, structural economic inequities, drug-resistant strains, and increasing rates of intravenous drug use, a growing risk factor for hepatitis B and C transmission in Mexico [11,12,13,14]. Simultaneously, Mexico is experiencing a nutritional transition, resulting in high rates of overweight and obesity of over 75% among adults. This shift has contributed to a heightened burden of cardiometabolic diseases, including MASLD, even in normal-weight individuals [15,16,17]. Mexicans are considered a population with elevated genetic and metabolic susceptibility to MASLD [18]. Shen et al. (2024) reported a disproportionately high prevalence of MASLD among Mexican Americans in the United States [19]. In Mexico, the estimated national prevalence of MASLD reached 41.3% in 2023 [20]. Additionally, recent findings from our research group revealed that among metabolically at-risk individuals, 57% were at risk for MASH. Notably, 67.8% of these patients, including 46% of those with normal weight, exhibited liver fibrosis or histopathological evidence of liver damage [16].

As a result, cardiometabolic conditions, including type 2 diabetes and liver disease, stand among the leading causes of morbidity and mortality nationwide [21]. Notably, chronic viral and metabolic-derived diseases require medical and nutritional counseling to reduce the risk of long-term liver damage. However, there are no official nutritional guidelines for managing viral hepatitis B and C, nor are there updated MASLD/MASH recommendations tailored to the Mexican population [22,23,24,25]. In this sense, earlier studies have highlighted the antiviral and hepatoprotective properties of specific nutrients and bioactive compounds found in the Traditional Mexican diet, which have been explored through the regionalized and genome-based dietary model known as the GENOMEX diet. This intervention combines traditional Mexican food staples with genomic, cultural, and clinical profiling to offer genome-based nutritional strategies. Previous studies have shown that the GENOMEX diet improves metabolic risk factors in individuals with obesity and dyslipidemia, and that its bioactive components exhibit antiviral activity against hepatitis B and C viruses [26,27,28,29,30]. Thus, searching for alternative strategies based on the population’s genomic and food culture background could potentially contribute to offering a culturally relevant, evidence-based framework for improving liver health in Mexico.

Genomic research has provided innovative nutrigenetic and nutrigenomic approaches focusing on nutrients and bioactive compounds that activate or inhibit metabolic pathways [31]. Furthermore, bioinformatics enables the comprehensive integration of molecular interactions across the genomic landscape, revealing network characteristics intricately linked to the pathogenesis of diseases [32]. This study aims to conduct a literature-based review integrated with bioinformatics to identify the antiviral and hepatoprotective pathways activated by ingredients in the traditional Mexican diet, exploring their potential clinical implications for the prevention and management of liver diseases.

## 2. Materials and Methods

### 2.1. Literature Review

#### 2.1.1. Data Sources for the Literature Review

A literature review was conducted using PubMed and Google Scholar databases to identify relevant studies on bioactive compounds in Mexican foods and their effects on s HBV, HCV and MASLD. Studies were included if they met the following criteria: (i) they presented experimental, mechanistic, or clinical evidence on the effects of nutrients or bioactive compounds; (ii) they focused specifically on traditional Mexican foods or their derived ingredients; and (iii) they addressed outcomes related to HBV, HCV, or MASLD. Both human studies and experimental models (in vitro, in vivo, or in silico) were eligible. Studies were excluded if they did not investigate Mexican foods or their nutrients or bioactives, lacked relevant experimental data, addressed unrelated health conditions, were duplicates, or were not original research articles (e.g., reviews, commentaries, or editorials). Only peer-reviewed original articles published in English or Spanish were considered. The review and screening process was independently performed by two researchers with expertise in nutrigenomics and liver disease. Disagreements were resolved through discussion and consensus.

The search strategy employed Medical Subject Headings (MeSH) terms to establish a systematic framework, refined through Boolean operator combinations in PubMed and Google Scholar. This approach prioritized the identification of high-quality, peer-reviewed studies. The literature review was categorized into two sections to ensure analytical clarity: (1) antiviral nutrients and bioactive compounds and (2) anti-MASLD nutrients and bioactive compounds. This approach facilitated a systemic integration of therapeutic mechanisms specific to viral hepatitis and metabolic liver disease.

#### 2.1.2. Search Strategy for Antiviral Nutrients

For the identification of antiviral nutrients, two independent literature searches were performed using Google Scholar and PubMed, each with a tailored Boolean operator. In Google Scholar, the Boolean operator used was as follows: (“Hepatitis B” OR “Hepatitis C” OR “Viral Hepatitis” OR “Hepatitis B Virus” OR “Hepatitis C Virus” OR “Hepatitis, Viral, Chronic”) AND (“Diet” OR “Nutrients” OR “Bioactive compounds”) AND (“Mexican diet” OR “Traditional Mexican foods”) AND (“clinical trial” OR “randomized controlled trial” OR “In silico” OR “In vivo” OR “In vitro”). For PubMed, the same search terms were applied, except that the category (“Mexican diet” OR “Traditional Mexican foods”) was replaced by (“Mexico”). The PubMed search returned four results, while the Google Scholar search yielded 69 results. After applying the predefined inclusion and exclusion criteria, a total of 27 articles were selected for qualitative analysis.

#### 2.1.3. Search Strategy for Anti-MASLD Nutrients

For the identification of Mexican nutrients and bioactive compounds with therapeutic potential against MASLD, two independent literature searches were performed using Google Scholar and PubMed, each with a tailored Boolean operator. In Google Scholar, the Boolean operator used was as follows: (“MASLD” OR “Non-alcoholic Fatty Liver Disease” OR “Fatty Liver”) AND (“Insulin Resistance” OR “Diabetes mellitus” OR “Type 2 Diabetes”) AND (“Diet” OR “Nutrients” OR “Bioactive compounds”) AND (“Mexican diet” OR “Traditional Mexican foods”) AND (“clinical trial” OR “randomized controlled trial” OR “In silico” OR “In vivo” OR “In vitro”). For PubMed, the same search terms were applied, except that the category (“Mexican diet” OR “Traditional Mexican foods”) was replaced by (“Mexico”). The PubMed search returned six results, while the Google Scholar search yielded 134 results. After applying the predefined inclusion and exclusion criteria, a total of 41 articles were selected for qualitative analysis.

#### 2.1.4. Classification of the Biological Effect of Antiviral and Anti-MASLD Nutrients

The biological effects of antiviral and anti-MASLD nutrients retrieved from the literature review were categorized into distinct functional groups: antiviral activity, which included the inhibition of viral entry and replication, as well as the enhancement of immune responses; antioxidants and anti-inflammatory properties, such as the upregulation of endogenous antioxidant systems and the reduction in inflammatory markers; metabolic effects, which included hypoglycemic activity (lowering blood glucose levels), insulin-sensitizing properties (enhancing insulin sensitivity and reducing insulin resistance), and hypolipidemic effects (decreasing total cholesterol, triglycerides, and VLDL while increasing HDL levels); anthropometric improvements through reducing body fat percentage and body weight; anorexigenic effects by enhancing satiety via prebiotic activity (promoting microbiota eubiosis); hepatoprotective effects (modulating liver enzymes, steatosis, and fibrosis); and anti-carcinogenic properties (upregulating tumor suppressor genes).

### 2.2. Integrative Bioinformatic Analysis

#### 2.2.1. Selection of Ingredients and Nutrients

For the Integrative Bioinformatic Analysis, we prioritized nutrients and bioactive compounds from five traditional Mexican dietary staples, selected based on their documented biological effects in prior literature, to enhance the clarity of genomic data integration and visualization. Four of these ingredients (maize, beans, chili, and tomato) are core components of the *milpa* diet, a traditional agricultural and culinary system central to Mexican cuisine and ubiquitously consumed across the country [33]. Avocado was additionally included for its unique nutrient profile, particularly its anti-carcinogenic compounds such as manganese, potassium, vitamin E, and monounsaturated fatty acids, which are linked to hepatoprotective and metabolic benefits [34].

#### 2.2.2. Identification of Gene Interactions with Nutrients and Bioactive Compounds

Using the Comparative Toxicogenomic Database (CTD) (https://ctdbase.org/, accessed on 4 March 2025), we retrieved the interacting genes linked to the identified nutrients and bioactive compounds. The analysis was targeted to deploy the top 10 genes involved in viral hepatitis, immune response, inflammation, nutrient metabolism, antioxidant mechanisms, insulin signaling, fibrosis, apoptosis, and cancer-related pathways [35]. These top 10 genes were selected for each nutrient or bioactive compound based on the highest number of curated chemical–gene interactions reported in CTD, which reflects both their biological relevance and frequency of interaction. In CTD, “top interacting genes” refers to those genes with the most curated interactions with a given chemical, derived from manually curated evidence across vertebrate and invertebrate species in the published literature.

#### 2.2.3. Functional Enrichment Analysis

For the Functional Enrichment Analysis, we utilized the bioinformatics tools STRING (https://string-db.org/, accessed on 4 March 2025), DAVID (https://davidbioinformatics.nih.gov/ortholog.jsp, accessed on 4 March 2025), and Enrichr (https://maayanlab.cloud/Enrichr/, accessed on 4 March 2025) to identify KEGG pathways influenced by the top 10 genes (to optimize visualization and enhance the clarity and interpretability of genomic data integration) interacting with nutrients and bioactive compounds derived from the selected ingredients [36,37,38,39]. All enrichment tools used the Fisher exact test, based on the hypergeometric distribution, to evaluate gene overrepresentation. DAVID applied the EASE score, a conservative variant of the Fisher exact test. The results were filtered using False Discovery Rate (FDR) correction with the Benjamini-Hochberg method (−Log FDR), using the *Homo sapiens* whole genome as the reference background. Pathways with FDR-adjusted *p* values < 0.05 were considered statistically significant.

#### 2.2.4. Data Visualization of the Integrative Bioinformatic Analysis

For comprehensive data visualization, we employed three complementary approaches, all conducted in RStudio (version 4.3.1) [40]. First, a heatmap was generated to display the frequency of co-occurrence between food sources and their associated genes, based on interactions retrieved from the CTD database. The interaction matrix was constructed by counting gene–food pairings and reshaping the data using tidyr [41] and reshape2 packages [42]. The heatmap was plotted using the ggplot2 package [43], with a visually optimized color scale from RColorBrewer [44] to enhance contrast and legibility.

Second, we generated a five-level Sankey diagram using the ggsankey package [45] to visualize the relationships between food sources, nutrients, genes, KEGG pathways (https://www.genome.jp/kegg/, accessed on 4 March 2025) [39], and potential liver diseases. A custom color palette from RColorBrewer was used to distinguish node categories.

Finally, we constructed an undirected, weighted co-occurrence network plot to illustrate integrated associations between foods, nutrients, genes, KEGG pathways (https://www.genome.jp/kegg/, accessed on 4 March 2025), and liver diseases. The network was created using the igraph and ggraph packages [35,36,46]. Input data were reshaped with tidyr, and edge weights were computed based on co-occurrence frequencies between nodes and disease terms. Node-representing distinct biological or nutritional entities were color-coded by category and scaled in size according to degree centrality. Edge thickness reflects interaction frequency. The layout was computed using the Fruchterman–Reingold force-directed algorithm to cluster highly connected nodes visually. All nodes were labeled for clarity and interpretability.

Figure 1 summarizes the workflow for identifying antiviral and anti-MASLD nutrients/bioactive compounds through a literature review, followed by the computational pipeline used in the Integrative Bioinformatic Analysis.

## 3. Results

### 3.1. Literature Review of Nutrients in the Mexican Diet Against HBV, HCV, and MASLD

We categorized the literature review into two thematic sections: antiviral nutrients (Appendix A; References [47,48,49,50,51,52,53,54,55,56,57,58,59,60,61,62,63,64,65,66,67,68,69,70,71,72,73,74]) and anti-MASLD nutrients/bioactive compounds (Appendix A; References [75,76,77,78,79,80,81,82,83,84,85,86,87,88,89,90,91,92,93,94,95,96,97,98,99,100,101,102,103,104,105,106,107,108,109,110,111,112,113,114,115,116,117,118,119]) to improve clarity and organization. The initial search yielded 73 scientific papers for antiviral nutrients (4 from PubMed and 69 from Google Scholar), of which 27 articles were selected based on the predefined methodology. Similarly, for anti-MASLD nutrients, the initial search retrieved 140 scientific articles (6 from PubMed and 134 from Google Scholar), with 41 studies ultimately chosen according to the established criteria.

Figure 2 illustrates the biological effects of traditional Mexican ingredients, proportionally categorized by their therapeutic contributions. These effects underscore the potential of regional nutrients and bioactive compounds targeting viral hepatitis, MASLD, and preventing associated chronic liver damage and progression to cirrhosis and HCC [28,29].

#### 3.1.1. Antiviral Nutrients

As illustrated in Figure 2 and depicted in Appendix A several nutrients have potential anti-HBV and anti-HCV activity (14.9%) which been described to disrupt HBV replication and mitigate its oncogenic progression. For instance, resveratrol in Mexican peanuts has shown inhibitory effects on HBV replication and tumor volume reduction in HBV-induced HCC in both in vitro and in vivo models [28,47]. Vitamin E, found in avocado and sunflower seeds, has demonstrated a reduction in HBV DNA levels and inflammatory cytokines in randomized controlled trials [48,49]. Lactoferrin, a bioactive protein in milk, blocks HBV entry into hepatocytes through HBsAg binding, as shown in vitro [50,51]. Similarly, selenium, present in tuna and sardines, has been shown in clinical trials to activate p53 and reduce HBV transcription, lowering HCC risk [52].

Furthermore, curcumin—derived from turmeric—has been shown, in vitro and in vivo, to suppress HBV transcription and replication, lower cccDNA levels, and inhibit NF-κB–mediated inflammatory signaling [53,54]. Luteolin-7-*O*-glucoside, a compound found in Mexican oregano, decreases HBV RNA and DNA levels, inhibits the secretion of HBsAg, and exhibits antioxidant and immunomodulatory activity in vitro [55]. Likewise, *Moringa oleifera* extracts have been shown, in vitro, to reduce cccDNA levels, inhibit NFκB signaling pathways, and exert antifibrotic effects. In addition, chlorogenic acid—present in sunflower seeds and coffee—has been shown in vitro to lower the secretion of HBsAg and HBeAg and slow the progression of liver fibrosis [56,57,58,59,60].

Lastly, epigallocatechin-3-gallate (EGCG) from cacao degrades sodium taurocholate cotransporting polypeptide (NTCP) receptors, which are key entry points for HBV, and stimulates the formation of autophagosomes that help eliminate viral components, as demonstrated in vitro [61,62,63,64]. Together, these Mexican ingredients provide a complementary and potentially cost-effective strategy to counteract HBV infection, mitigate liver injury, and reduce the risk of progression to cirrhosis or HCC.

On the other hand, various foods supply nutrients that may directly or indirectly inhibit HCV replication and mitigate liver damage, as summarized in Appendix A. Docosahexaenoic acid (DHA), found in fish and seafood such as sardines, has been shown in vitro, to counteract lipid disruptions induced by HCV core proteins, thereby reducing viral replication. Likewise, eicosapentaenoic acid (EPA), also derived from fish, exhibits anti-inflammatory properties and impedes viral propagation in similar experimental settings [65,66]. Gallic acid in cloves and oregano diminishes HCV expression through antioxidant mechanisms, as demonstrated in vitro [67]. Meanwhile, vitamin E—found in sunflower seeds and avocado—has been shown to reduce ALT levels and inflammatory markers in a randomized controlled trial [68].

Other micronutrients, such as vitamin A (from liver and carrots) and vitamin D3 (from fish and eggs), support interferon-mediated antiviral pathways and inhibit viral replication, as demonstrated in vitro and prospective cohort studies, respectively [69,70,71]. Vitamin B12 (from green leafy vegetables), dietary iron (from marjoram, leafy greens, and beans), and zinc (from agave and sesame seeds) have all demonstrated antiviral activity against HCV in vitro. Specifically, B12 targets internal ribosome entry sites [72], iron inhibits HCV polymerase activity [73], and zinc interferes with RNA synthesis to suppress replication [74]. These nutrients may be further explored as part of nutritional therapeutic strategies to manage HCV infections in the Mexican population.

#### 3.1.2. Anti-MASLD Effect

As illustrated in Figure 2 and Appendix A, this literature review revealed the potential therapeutic biological effects of the traditional Mexican ingredients, including hypoglycemic (14.9%), insulin-sensitizing (13.8%), antioxidant (11.7%), lipid-lowering (10.6%), anti-inflammatory, and anthropometric improvements (7.4%). In a minor proportion, anorexigenic (3.2%), anti-carcinogenic (3.2%), and prebiotic (1.1%) effects were observed. All these effects potentially mitigate liver damage progression to MASH, cirrhosis, and HCC.

The following section outlines the principal therapeutic potentials of nutrients and bioactive compounds derived from *Zea mays* (maize), *Phaseolus vulgaris* (common bean), *Solanum lycopersicum* (tomato), *Capsicum annuum* (chili pepper), and *Persea americana* (avocado). These species hold profound culinary and cultural significance in Mexican cuisine, with accumulating scientific evidence underscoring their roles in modulating metabolic, inflammatory, and oxidative pathways. Appendix A expands this analysis by including additional Mexican foods, including *Theobroma cacao* (cacao), *Pachyrhizus erosus* (jicama), *Agave* spp. (agave), *Psidium guajava* (guayaba), *Opuntia Ficus Indica* (nopal), *Opuntia Robusta fruit* (prickly pear fruit), *Opuntia cochenillifera* (Nopal Cladodes), *Carica papaya* (papaya), *Arachis hypogaea* (peanut), *Salvia hispanica* (chia), *Cucurbita maxima* (pumpkin seeds), *Helianthus annuus* (sunflower seeds), *Linum usitatissimum* (flaxseed), *Carya Illinoensis* (pecan), *Amaranthus* spp. (quelites), and *Portulaca oleracea* (purslane).

##### Hypoglycemic and Insulin-Sensitizing Effect

The regulation of blood glucose levels and insulin sensitivity is fundamental to metabolic health, particularly for type 2 diabetes mellitus, MASLD, and MASH. Several traditional Mexican foods exhibit hypoglycemic and insulin-sensitizing properties, primarily through enhanced glucose uptake, modulation of insulin signaling pathways, inhibition of carbohydrate digestion, and regulation of metabolic enzymes [120,121].

Beans (*Phaseolus vulgaris*), a staple in Mexican cuisine, contain insoluble fiber, anthocyanins, α-amylase inhibitors, and lectins, all of which contribute to lowered fasting glucose levels, enhanced GLP-1 secretion, and improved glucose homeostasis. Additionally, beans slow carbohydrate digestion and absorption, reducing glycemic variability. These effects have been demonstrated through in vitro, in silico, and clinical randomized double blind controlled trials [31,90,91,95,96,97]. Cacao (*Theobroma cacao*) enhances GLP-1 expression and insulin secretion through its procyanidins and epicatechins, contributing to improved postprandial glucose control and enhanced insulin sensitivity in randomized control trials and in vitro essays [85,88].

Avocado (*Persea americana*), a rich source of monounsaturated fatty acids (MUFAs), phytosterols, perseitol, and avocatin B, has been associated with improved glucose utilization, lower fasting blood glucose levels, and modulation of insulin signaling pathways—such as AKT phosphorylation [115,116]. These effects have been demonstrated both in vivo and in randomized, double-blind, controlled clinical trials [98,99,100]. Chia seeds (*Salvia hispanica* L.), rich in α-linolenic acid (ALA), fiber, quercetin, and myricetin, have been shown to promote GLUT-4 translocation in muscle tissue and improve insulin sensitivity. These mechanisms contribute to better fasting glucose control and enhanced metabolic efficiency, as shown in vivo and in randomized, double-blind clinical trials [101,102,103].

Tomato (*Solanum lycopersicum*) also exhibits hypoglycemic effects, primarily attributed to its fiber, lycopene, and β-carotene content. These bioactive compounds help slow carbohydrate absorption, resulting in reduced fasting glucose levels and improved postprandial glycemic control, as demonstrated in vivo, in randomized controlled trials, and in case–control studies [117,118,119]. Chili (*Capsicum* spp.) exerts insulin-sensitizing effects through its active compound, capsaicin, which promotes glucose uptake in muscle cells by activating AMPK and p38 MAPK pathways. Furthermore, chili consumption has been associated with increased insulin secretion, supporting glycemic regulation, as demonstrated in vitro assays and crossover clinical trials [104,105,106,107].

##### Lipid-Lowering Effect

Dyslipidemia, characterized by elevated triglycerides, total cholesterol, and low-density lipoprotein cholesterol (LDL-C), along with reduced high-density lipoprotein cholesterol (HDL-C), is a key contributor to MASLD, MASH, and cardiovascular disease [116,122]. Several traditional Mexican foods possess bioactive compounds that regulate lipid metabolism by promoting fatty acid oxidation, inhibiting cholesterol synthesis, enhancing lipid transport, and reducing hepatic lipid accumulation.

Tomato (*Solanum Lycopersicum*), a rich source of lycopene, β-carotene, and dietary fiber, has been associated with reductions in triglycerides, total cholesterol, and LDL-C. Lycopene functions as a peroxisome proliferator-activated receptor gamma (PPARγ) inhibitor, modulating lipid metabolism by suppressing adipogenesis and enhancing lipoprotein lipase (LPL) activity, thereby promoting triglyceride clearance, as demonstrated in vivo and in clinical trials [117,118,119]. Avocado (*Persea americana*), another widely consumed food, is an excellent source of monounsaturated fatty acids, phytosterols, and soluble fiber. These compounds reduce total cholesterol, LDL-C, and small dense LDL particles while increasing HDL-C. Avocado also inhibits the activity of cholesterol ester transfer protein (CETP), which plays a role in lipoprotein remodeling and lipid transport [98,115,116].

Blue corn (*Zea mays*), rich in anthocyanins and polyphenols, has been shown to lower total cholesterol, triglycerides, and LDL-C, while enhancing HDL-C synthesis. These bioactive compounds modulate key cholesterol transporters, including ATP-binding cassette transporter A1 (ABCA1) and liver X receptor alpha (LXRα), facilitating cholesterol efflux and improving lipid homeostasis. These effects have been demonstrated through in silico, in vitro, and in vivo studies [31,89,90,91,92,93,94,123]. Beans (*Phaseolus vulgaris*) have been associated with reductions in plasma triglycerides, total cholesterol, and LDL-C levels. These lipid-lowering effects are primarily attributed to their high content of insoluble fiber, which facilitates bile acid excretion, and their capacity to modulate lipid absorption in the intestine. These findings have been supported by evidence from in vitro, in silico, and randomized double-blind clinical trials [90,91,95,96,97,124].

##### Antioxidant and Anti-Inflammatory Effects

Chronic oxidative stress and inflammation are central to the progression of MASLD, MASH, insulin resistance, and cardiovascular diseases. Oxidative stress occurs when there is an imbalance between the production of reactive oxygen species (ROS) and the body’s antioxidant defense mechanisms, resulting in lipid peroxidation, mitochondrial dysfunction, and DNA damage. Simultaneously, persistent low-grade inflammation, driven by cytokine dysregulation, contributes to insulin resistance, fibrosis, and hepatic injury [116,125]. Several traditional Mexican foods contain bioactive compounds that exert antioxidant and anti-inflammatory effects by neutralizing free radicals, enhancing the activity of endogenous antioxidant enzymes, and suppressing pro-inflammatory pathways.

The anthocyanins and phenolic compounds found in blue corn (*Zea mays*) have similarly been linked to reductions in oxidative damage, as evidenced by decreased MDA levels and increased hepatic superoxide dismutase (SOD1) expression [31,89,90,91,92,93,94,123]. These findings suggest a protective role against oxidative stress-related metabolic dysfunction. Beyond their antioxidant activity, several traditional Mexican foods also exert anti-inflammatory effects by modulating key cytokine pathways. Tomato (*Solanum lycopersicum*), rich in lycopene and β-carotene, has been shown to reduce levels of pro-inflammatory cytokines—particularly tumor necrosis factor-alpha (TNF-α)—in clinical trials [117,118,119]. Chili (*Capsicum* spp.), through capsaicin, a bioactive compound with anti-inflammatory properties, inhibits nuclear factor-kappa B (NF-κB) signaling. By downregulating the expression of pro-inflammatory mediators, capsaicin helps mitigate chronic inflammation associated with metabolic disease [104,105,106,107]. These effects reduce systemic inflammation, which is particularly relevant in conditions such as MASLD and insulin resistance.

##### Hepatoprotective and Potential Anti-Carcinogenic Effect

Liver health is crucial for maintaining metabolic homeostasis, and disruptions in hepatic function contribute to MASLD/MASH, fibrosis, and cirrhosis [116,125]. Several traditional Mexican foods contain bioactive compounds that exert hepatoprotective effects by reducing liver enzyme levels, preventing fibrosis, regulating lipid metabolism, and mitigating oxidative damage to hepatocytes.

Experimental studies have shown that blue corn (*Zea mays*, white and blue varieties) exerts hepatoprotective effects by reducing liver weight, hepatic steatosis, and inflammatory foci. Its anthocyanins and polyphenols mitigate lipid accumulation and oxidative damage in hepatocytes, thereby preventing liver injury associated with metabolic dysfunction [31,89,90,91,92,93,94,123]. In parallel, *Opuntia robusta* fruit (prickly pear), a traditional Mexican cactus fruit rich in betacyanins and betalains, has demonstrated potent hepatoprotective and anti-carcinogenic properties. In vivo models revealed significant reductions in serum AST and ALT levels, as well as decreased caspase-3 activity, indicating attenuation of liver damage and apoptosis. Notably, *Opuntia* consumption led to increased hepatic *P53* expression, highlighting its potential role in cancer prevention [86]. Together, these findings underscore the relevance of native Mexican foods as promising dietary strategies for preserving liver health and preventing liver-related carcinogenesis.

##### Gut Microbiota Modulation by Mexican Foods on Liver Health

Recent evidence highlights the importance of gut microbiota modulation in the prevention and management of MASLD through traditional dietary components. Prebiotic fibers from culturally relevant Mexican foods—such as beans, chia seeds, maize, quelites, and avocado—promote the growth of beneficial bacteria, stimulate SCFA production, and enhance gut barrier integrity, thereby reducing endotoxin translocation and liver inflammation. Clinical and preclinical studies consistently report that prebiotics help decrease hepatic steatosis, inflammation, fibrosis, and liver enzyme abnormalities [126,127,128,129,130].

Additionally, omega-3 polyunsaturated fatty acids (PUFAs) found in fish, chia, and avocado further support a hepatoprotective microbiota profile by enriching SCFA-producing taxa such as *Lachnospiraceae*, *Prevotella*, *Roseburia*, and *Ruminococcus* [131]. In contrast, high intake of saturated fatty acids (SFAs)—characteristic of ultra-processed foods in the obesogenic Mexican environment—has been linked to reduced microbial diversity and a lower abundance of fiber-degrading bacteria like *Acetivibrio cellulolyticus* and *Clostridium* spp., contributing to hepatic steatosis and metabolic dysfunction [132]. These findings underscore the value of traditional Mexican diets, rich in fiber and PUFAs, as a culturally relevant strategy to counteract the liver-related consequences of modern dietary patterns.

Findings from this review highlight the therapeutic potential of traditional Mexican foods in delaying MASLD progression through multiple mechanisms. Their bioactive compounds exert hypoglycemic, lipid-lowering, antioxidant, anti-inflammatory, hepatoprotective, and prebiotic effects, while also modulating the gut–liver axis. Collectively, these effects underscore the value of culturally relevant dietary strategies rooted in traditional Mexican cuisine to prevent chronic liver injury and its progression to MASH, cirrhosis, and hepatocellular carcinoma.

### 3.2. Integrative Bioinformatic Analysis

#### 3.2.1. Gene–Nutrient and Bioactive Compound Interactions

The criteria for ingredient selection are detailed in Section 2. Figure 3 illustrates gene interaction networks for maize, beans, chili, tomato, and avocado, emphasizing critical molecular pathways. Prominent genes showing frequent interactions—including *TNF*, *IL6*, *IL1B*, and *PTGS2*—are central to immune system regulation. Beans emerged as the ingredient with the most robust gene interactions, particularly involving genes associated with immune response (*TNF*, *IL-6*, and *IL-1β)*, apoptosis regulation *(BAX*, *BCL-2*), oxidative stress response (*NFE2L2*), and fibrotic processes (*TGFB1*).

Avocado displays a moderate but broad impact, influencing genes related to immune response and apoptosis, including *TNF*, *IFNG*, *BAX*, and *CASP3*. Maize primarily interacts with genes involved in immune modulation and apoptotic pathways, such as *TNF*, *PTGS2*, *CASP3*, *IL1B*, and *IL6*. Tomatoes primarily interact with the regulation of apoptosis and oxidative stress, as well as with *BAX*, *CASP3*, and *CAT*. Chili exhibits a distinct interaction pattern, influencing tumor suppression, cell proliferation, survival, and apoptosis-related genes, including *TP53*, *CASP3*, and *MAPK1*/*2*.

#### 3.2.2. Functional Enrichment Analysis Visualization

Functional Enrichment Analysis, illustrated in the Sankey diagram (Figure 4), reveals a complex network of interactions between bioactive compounds from these traditional Mexican foods, their target genes, and key KEGG pathways linked to liver diseases, viral hepatitis, MASLD, and HCC.

The Functional Enrichment Analysis highlights that bioactive compounds from maize, beans, chili, tomato, and avocado exert pleiotropic effects on key metabolic, antioxidant, immune, fibrogenic, and carcinogenic pathways (Figure 4). These compounds modulate the expression of genes involved in lipid metabolism (*PPARA*, *IRS1*, and *FASN*), inflammatory signaling (*TNF*, *IL-6*, *RELA*, *IFN-γ*, and *IL-1β*), and apoptotic regulation (*BAX*, *BCL-2*, and *CASP3*). The main interacting KEGG pathways were related to carcinogenesis (*P53*, cancer pathways, and hepatocellular carcinoma), viral hepatitis, and metabolic dysfunction (diabetes mellitus type 1 and 2, *HIF1A* signaling, peroxisomes, and Lipolysis). These findings align with the evidence from the literature review, where maize, beans, chili, tomato, and avocado were shown to improve insulin sensitivity and have hypoglycemic, lipid-lowering, antioxidant, and anti-inflammatory effects.

The network plot (Figure 5) illustrates the complex relationship between Mexican foods and key biological processes associated with viral hepatitis, MASLD, and HCC. In the context of viral hepatitis, the predominant genes and pathways identified were those involved in modulating the immune response. For MASLD, the most relevant genetic interactions and pathways were linked to metabolic dysfunction, oxidative stress regulation, and inflammation. In the case of HCC, the analysis highlighted tumor suppression pathways and genes associated with apoptosis, cell proliferation, and antioxidant defense mechanisms.

Additionally, the network analysis revealed that avocado, chili, tomato, maize, and beans were all interconnected with viral hepatitis, MASLD, and HCC, though the bioactive nutrients driving these interactions varied. Specifically, vitamin E was a common link between viral hepatitis and HCC, while potassium was associated with both viral hepatitis and MASLD. Similarly, lycopene and anthocyanins were shared between MASLD and HCC, suggesting their dual role in metabolic and oncogenic pathways. Notably, folate, monounsaturated fatty acids (MUFAs) and phytosterols were exclusively associated with MASLD, indicating their potential role in metabolic regulation and liver health.

Overall, the convergence of data from genomic enrichment and nutritional studies highlights the potential of the traditional Mexican diet as a nutritional strategy for preventing liver disease. The combination of diverse bioactive compounds within this dietary pattern provides synergistic effects that modulate key molecular pathways involved in metabolic regulation, antioxidation, inflammation, fibrosis, and carcinogenesis.

#### 3.2.3. Integration of Nutrigenomic Interactions of Traditional Mexican Foods

Figure 6 illustrates the complex molecular interplay between viral infections (HBV and HCV), metabolic dysfunction, and HCC, highlighting the modulatory role of bioactive compounds derived from traditional Mexican foods as identified through enrichment analysis.

These interactions span key pathways related to immune evasion, insulin resistance, oxidative stress, inflammation, uncontrolled apoptosis, and tumor proliferation, all of which are central to the pathogenesis of chronic liver disease. Viral hepatitis generates liver damage by hijacking host immune signaling. HBV and HCV evade innate immunity by suppressing *RIG-I*, *TLR3*, and *NFκB* pathways, leading to persistent inflammation and hepatocyte injury [28,29]. The viral proteins HBsAg, HBx, and NS5A modulate key cellular pathways, including *ERK*, *JNK*, and *mTOR*, thereby contributing to viral persistence and oncogenesis [3,133,134,135]. However, bioactive compounds such as anthocyanins, polyphenols, flavonoids, saponins, and proanthocyanidins can interfere with these mechanisms, potentially enhancing antiviral immune responses while mitigating inflammation [136,137,138,139,140].

Insulin signaling (*INS/INSR*) is impaired by pro-inflammatory cytokines, such as IL-6 and TNF-α, which promote hepatic lipid accumulation and fibrosis [141,142]. Nutrients such as fiber, capsaicin, potassium, manganese, and vitamin E may counteract these effects by improving insulin sensitivity and modulating inflammatory pathways [143,144,145,146,147]. Additionally, monounsaturated fatty acids (MUFAs) may activate *PPARα*, a key regulator of lipid metabolism and generator of peroxisomes, thereby reducing hepatic steatosis and preventing metabolic stress [148,149].

Oxidative stress plays a central role in the progression of liver disease, with excessive ROS production depleting *NRF2*-mediated antioxidant responses [150]. Excessive oxidative stress stimulates *TGF-β* signaling, promoting fibrosis and chronic inflammation through the actions of IL-6 and TNF-α [151,152,153,154]. Bioactive molecules, such as ferulic acid, lycopene, vitamin E, and proanthocyanidins, may help neutralize oxidative damage, enhance hepatic antioxidant defenses (HMOX and *NQO1*), and prevent fibrotic remodeling. These compounds contribute to hepatoprotection by maintaining redox homeostasis and inhibiting fibrogenic pathways [155,156,157,158].

Apoptosis regulation is also crucial in liver pathology, as it balances cell survival and programmed cell death. Dysregulation in apoptotic pathways, characterized by increased BAX, CASP3, and Jun expression, leads to excessive hepatocyte loss. In contrast, oncogenic signals, such as HBx, suppress p53, thereby fostering genomic instability and promoting tumor progression [159,160,161,162,163]. Lycopene, β-carotene, polyphenols, saponins, and flavonoids may restore apoptotic balance by preventing mitochondrial dysfunction, reducing oncogenic mutations, and promoting controlled cell death [34,156,164,165,166].

HCC progression is partly fueled by the activation of mTOR and ERK1/2, which stimulate cell proliferation and survival [167]. The suppression of tumor suppressors (*p21/p53*) and the overexpression of *CCND1* (Cyclin D1) accelerate oncogenic transformation [34,159,168]. However, dietary bioactive compounds such as capsaicin, polyphenols, manganese, and β-carotene exert antiproliferative effects, potentially inhibiting inflammatory mediators (TNF-α, CASP3), reducing pro-tumorigenic signals, and regulating autophagy, thereby limiting HCC progression [31,104,105,106,107].

## 4. Discussion

The gene–nutrient interactions of nutrients and bioactive compounds derived from the traditional Mexican foods analyzed in this study have clinical implications for preventing and managing HBV, HCV, MASLD, and ultimately HCC. The Integrative Bioinformatic Analysis revealed that food-derived molecules regulating key pathways linked to liver disease pathogenesis could serve as a basis for implementing therapeutic strategies to enhance synergistic outcomes when combined with antiviral therapies. Such integrative approaches may diminish viral persistence, alleviate insulin resistance, suppress chronic inflammatory cascades, and attenuate fibrotic progression and HCC risk. Therefore, by building on prior evidence from the literature review (summarized in Figure 2 and Appendix A) and the Integrative Bioinformatics Analysis, we propose nutrigenomic culinary recommendations rooted in traditional Mexican cuisine to address viral hepatitis, MASLD, and HCC. This framework bridges ancestral dietary patterns with precision nutrition to mitigate liver diseases.

Figure 7 presents a comprehensive nutrigenomic framework that integrates the antiviral, anti-MASLD, and anti-HCC capacities of staple Mexican foods, illustrating how their bioactive compounds can be combined into culturally relevant dishes to enhance liver health benefits. On the upper left side, beans and maize are highlighted for their antioxidant, anti-inflammatory, and metabolic-regulating effects, including reductions in oxidative stress, improved lipid metabolism, and modulation of inflammatory cytokines. In the upper right section, tomatoes are notable for their β-carotene and lycopene, which confer anti-carcinogenic activity by inhibiting cell proliferation and promoting apoptosis. Avocados, on the other hand, deliver potassium, manganese, vitamin E, and MUFAs, conferring both hepatoprotective and anti-carcinogenic benefits. Additionally, chili (*Capsicum* spp.) provides capsaicin, which has been shown to exert antiviral and tumor-suppressing actions.

In the lower section, Figure 7 links virus-specific nutrients (for both HBV and HCV) to their immunomodulatory and antiviral mechanisms, including EPA, DHA, vitamin A, E, and D3, as well as B12, iron, zinc, resveratrol, lactoferrin, selenium, curcumin, luteolin, and moringa extracts. Thus, this integrative model illustrates not only how these traditional Mexican foods target overlapping molecular pathways that underlie viral hepatitis, metabolic dysfunction, and HCC but also how they can be combined into whole-food dishes that align with both ancestral dietary practices and modern nutrigenomic insights for optimal liver health.

HBV and HCV remain significant global contributors to liver disease, ranging from chronic hepatitis to cirrhosis and HCC [1,169]. These viruses, though adaptive signatures in their genome and replication strategies, share common mechanisms of liver damage, including immune evasion, metabolic disruption, and the promotion of chronic inflammation, fibrosis, and carcinogenesis. Given the central role of immune evasion, metabolic disruption, and chronic inflammation in the pathogenesis of HBV and HCV, the bioactive compounds in the traditional Mexican diet emerge as potential modulators of these mechanisms.

For viral hepatitis, traditional Mexican ingredients like avocado, sunflower seeds, oregano, sardines, and leafy greens, such as purslane or amaranth leaves, provide bioactive compounds that help counteract the infection. Vitamin E (avocado, sunflower seeds) blocks HBV entry (NTCP receptor) and reduces cccDNA. Luteolin-7-*O*-glucoside (oregano) and curcumin suppress NFκB inflammation, while EGCG (cocoa) enhances viral clearance. Selenium (sardines, tuna) activates p53 for DNA repair, and DHA inhibits HCV polymerase. Vitamin B12 (leafy greens) disrupts HCV replication. A traditional antiviral dish could include grilled sardines or seasonal Mexican fish with an oregano rub, accompanied by guacamole salsa (rich in vitamin E) and a stew of leafy greens, served with *agua fresca* (fruit-flavored waters) infused with moringa, lime, and chia seeds. This meal supports the immune system, reduces viral replication, and helps control hepatic inflammation, providing a dietary approach to support antiviral therapies.

MASLD has become a significant cause of chronic liver disease and liver transplantation globally [170], driven mainly by an epidemiological shift linked to Westernized lifestyles [171,172,173]. This nutritional transition, characterized by a decline in the traditional diet rich in maize, beans, tomatoes, chili, avocado, and other staples, has been replaced by ultra-processed foods high in obesogens, sugars, and unhealthy fats, contributing to obesity, type 2 diabetes, and metabolic dysfunction-associated liver diseases [174,175].

The global nutrition transition intensifies ecosystem and human health dysfunctions by disrupting the delicate balance of gene–environment interactions [18,176,177,178,179]. Traditional diets have historically played a crucial role in supporting immune and metabolic health through millennial gene–nutrient adaptations [180]. However, this shift exemplifies an evolutionary mismatch: human metabolism co-evolved alongside nutrient-dense, ancestral diets, yet modern pathogenic dietary patterns dysregulate immune, antioxidant, and metabolic pathways [181,182,183,184]. This dysregulation fosters insulin resistance, oxidative stress, chronic inflammation, and progressive liver damage, further exacerbating the burden of metabolic and liver diseases [159,185]. In Mexico, approximately half of the adult population is affected by MASLD and faces a heightened risk of developing MASH, highlighting a critical public health issue [16,186].

For MASLD, incorporating Mexican ingredients such as jicama, beans, maize, agave, tomatoes, chili peppers, purslane, and chia seeds into traditional Mexican recipes can help slow the progression of metabolic dysfunction. Fiber and fructans (inulin and fructooligosaccharides) enhance insulin sensitivity and reduce hepatic steatosis, while anthocyanins and polyphenols support lipid metabolism and mitigate oxidative stress. MUFAs and phytosterols may lower LDL, raise HDL, and reduce liver fat. Additionally, micronutrients such as folate, vitamin E, potassium, and manganese help modulate inflammation and improve insulin responsiveness. Consequently, a traditional MASLD-focused dish could feature a white or blue variety of maize *sopes* with smashed black beans and chili–tomato sauce, accompanied by a purslane stew and fresh jicama slices and lime water sweetened with agave inulin and chia seeds for dessert.

For HCC, traditional Mexican ingredients such as peanuts, cocoa, sunflower seeds, tomatoes, blue corn, beans, chili, avocado, and international turmeric offer bioactive compounds to combat hepatocellular carcinogenesis. Resveratrol (peanuts) and curcumin (turmeric) induce apoptosis (via BAX/CASP3) and suppress oncogenic NFκB/mTOR pathways. Chlorogenic acid (found in sunflower seeds) and epicatechins (found in cocoa) inhibit tumor growth by blocking proliferation signals. Lycopene (tomato) and anthocyanins (blue corn and black beans) reduce oxidative DNA damage and modulate p53/BCL2 balance to promote controlled cell death, while β-carotene (tomato) inhibits cyclin D1 to curb uncontrolled division. Capsaicin (chili) exerts antiproliferative effects through MAPK pathway inhibition, and phytosterols (avocado) enhance TP53 tumor suppressor activity.

A traditional HCC-focused meal might be mole poblano (a sauce blending cocoa, different species of chili, and peanuts) served with maize tortillas and black beans, accompanied by a tomato–avocado salad, and a chia–cacao *atole* (corn-based drink) sweetened with agave inulin. This combination may potentiate synergistic anti-carcinogenic mechanisms, aligning with cultural and culinary practices, and offers a dietary strategy to complement HCC prevention.

These findings not only highlight the functional role of Mexican native foods in liver health but also support their inclusion in dietary guidelines aimed at reducing the burden of liver diseases in Mexico. However, despite these properties, clinical practice guidelines in Mexico continue to promote dietary patterns such as the Mediterranean and DASH diets, which are not culturally rooted in the Mexican context [22,23,24,25,187]. Notably, the 2023 Dietary Guidelines from the National Institute of Public Health mark a shift in advocating for a healthy Mexican diet that aligns with local ecosystems and cultural heritage [188].

Nevertheless, traditional Mexican staples like maize face economic and legal threats due to trade agreements, such as the United States–Mexico–Canada Agreement (USMCA or T-MEC), which is leading to the monopolization of genetically modified maize that resists pesticides and herbicides, thereby displacing native Mexican varieties, including blue maize [189,190,191]. This displacement undermines the nutritional diversity provided by autochthonous crops, which are rich in anthocyanins, polyphenols, and other bioactive compounds that may have shaped and sustained the metabolic health of the Mexican population [26,30,180].

Paradoxically, while avocado (*Persea americana*) was included due to its rich nutrient composition, which has demonstrated hepatoprotective properties and potential to prevent HCC [18,34,164], the increasing international demand for avocado has led to environmental degradation, deforestation, and unsustainable land use in Mexico. Moreover, economic profits remain heavily concentrated in American agribusiness, while the resulting regional and local environmental burdens increasingly affect the most vulnerable populations [192]. Therefore, this selection highlights the importance of striking a balance between nutritional benefits and sustainable agricultural practices, thereby preserving both ecosystem integrity and traditional food systems.

Although this study provides an integrative nutrigenomic perspective, several limitations should be acknowledged. As expected, the strength of evidence inherently varies across in silico, in vitro, and in vivo studies, and such evidence is not uniformly available for all traditional Mexican foods included in this analysis. Moreover, the literature may be biased toward compounds with previously documented biological activity, potentially overlooking lesser-known foods or nutrients. This work is not a systematic literature review; instead, it offers an exploratory, Integrative Bioinformatic Analysis based on curated literature sources and bioinformatic tools to generate nutrigenomic hypotheses. As such, the selection of studies may be subject to selection bias and limitations inherent to the databases used. While bioinformatic tools enable high-throughput hypothesis generation, they rely on existing curated datasets and may not fully capture the complexity of nutrient–gene interactions in physiological contexts. Therefore, in silico predictions should be interpreted with caution and validated through mechanistic studies, including experimental models and human clinical trials.

Additionally, many of the bioactive compounds identified have been studied primarily in isolated or supplement form, which may not accurately reflect their behavior when consumed as part of whole-food matrices. Their bioavailability, metabolic fate, and synergistic interactions could differ within traditional dietary contexts. Moreover, there is a notable lack of dietary intervention trials conducted specifically in Mexican populations. The broader implementation of GENOMEX diet strategies may also face important challenges, including persistent socioeconomic inequalities, shifts in agricultural practices, and the ongoing nutritional transition that is displacing traditional diets and contributing to the erosion of native crops such as maize. Nutrient–gene interactions are further shaped by a range of population-specific factors, including ancestry-related polymorphisms, epigenetic regulation, and microbiome diversity, which may modulate the biological effects of traditional foods. Since this study was developed in the context of the Mexican population, the extrapolation of findings to other populations with different genetic, environmental, and cultural backgrounds is limited. Collectively, these factors constrain the immediate translational applicability of the current findings and emphasize the need for rigorous, context-sensitive validation.

Despite these limitations, this study underscores several strengths. It offers a novel integrative approach, combining a narrative literature review with bioinformatic enrichment to elucidate nutrigenomic mechanisms linking traditional Mexican foods to pathways implicated in HBV, HCV, and MASLD. Rather than ranking foods, the analysis demonstrates how multi-component foods engage liver disease-related pathways, situating these mechanisms within a culturally grounded dietary pattern consumed for millennia by Native Mexican populations. By focusing on the genomic and cultural context of the Mexican population, this work highlights the potential of culturally and genomically tailored nutritional interventions. Notably, research groups in Mexico have already demonstrated the efficacy of this dietary pattern, with clinical studies reporting improvements in adherence, self-efficacy, and cardiometabolic health markers, further supporting its role in mitigating the burden of chronic liver and metabolic diseases [26,27,193].

While the GENOMEX diet model shares functional similarities with other heritage-based dietary patterns—such as the antioxidant and anti-inflammatory benefits of the Mediterranean and traditional Asian diets—its aim is not to replace or compete with these frameworks. Instead, the GENOMEX diet offers a population-specific, nutrigenomic approach grounded in the evolutionary, genomic, cultural, and ecological context of Mexico. Just as Mediterranean diets are best suited to Mediterranean populations, the GENOMEX diet highlights the health potential of native Mexican foods for the Mexican population. This model also serves as an invitation for other regions to reexamine their traditional food systems through a nutrigenomic lens, contributing to more inclusive, culturally adapted public health strategies. Hence, we invite other Latin American populations to reclaim their ancestral and traditional ingredients, resist nutritional and epistemological colonialism, and establish sovereign public health strategies designed specifically for their populations. These strategies should honor and integrate their cultural and genomic heritage, fostering health policies that are effective and deeply rooted in each nation’s identity, biodiversity, and history.

## 5. Conclusions

This study demonstrates that integrating traditional Mexican foods into dietary strategies provides a potential nutrigenomic solution to mitigate viral hepatitis B and C, MASLD, and reduce the risk of developing liver fibrosis and HCC in Mexican populations. This approach addresses urgent public health priorities such as curbing the cardiometabolic disease epidemic and alleviating the burden of viral hepatitis. This implication is particularly relevant in countries where antiviral therapies remain inaccessible due to cost barriers, genetic polymorphisms linked to treatment resistance and risk of cardiometabolic disease, and an escalating mismatch between the population’s evolutionary genetics and modern dietary patterns.

## Figures and Tables

**Figure 1 ijms-26-08977-f001:**
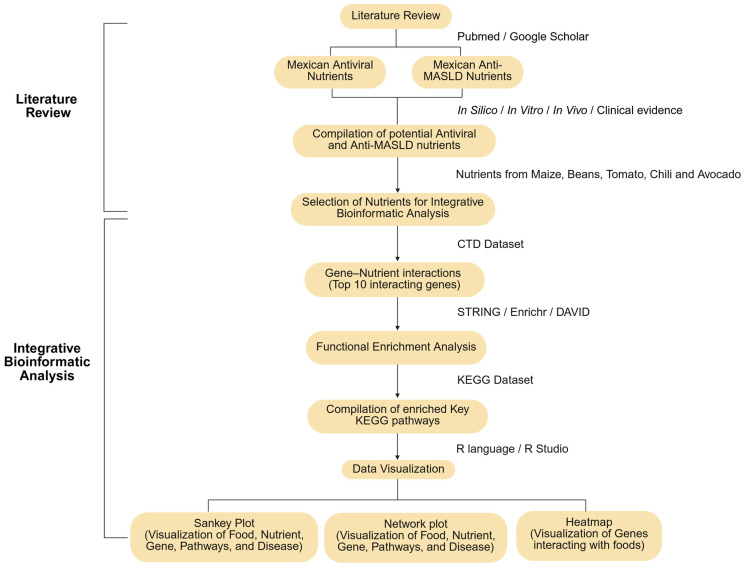
Schematic representation of the methodological workflow applied to conduct the literature review and Integrative Bioinformatic Analysis. Abbreviations: Anti-MASLD, anti-Metabolic dysfunction-Associated Steatotic Liver Disease; CTD, Comparative Toxicogenomic Database; STRING, Search Tool for the Retrieval of Interacting Genes/Proteins; DAVID, Database for Annotation, Visualization and Integrated Discovery; KEGG, Kyoto Encyclopedia of Genes and Genomes.

**Figure 2 ijms-26-08977-f002:**
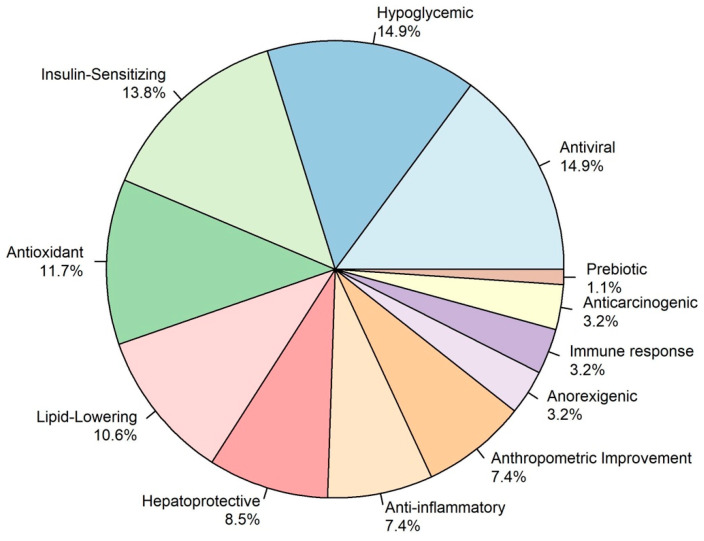
The effect of traditional Mexican ingredients on markers related to HBV, HCV, and MASLD, as found in the literature review. The pie chart illustrates the frequency distribution of identified biological effects from nutrients and bioactive compounds found in Mexican foods. Percentages represent the cumulative proportion of reported effects per food, as detailed in Appendix A. Abbreviations: HBV, Hepatitis B Virus; HCV, Hepatitis C Virus; MASLD, Metabolic Dysfunction-Associated Steatotic Liver Disease.

**Figure 3 ijms-26-08977-f003:**
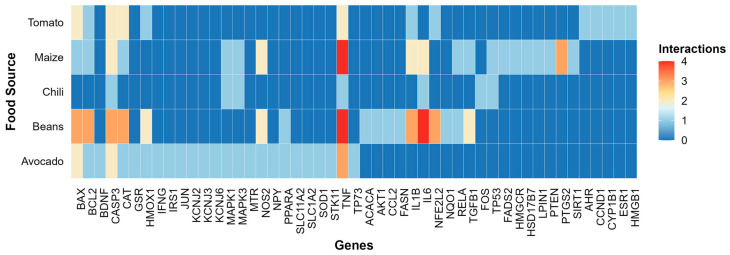
A heatmap illustrates the interactions between food sources and genes. The rows represent different food sources (tomato, maize, chili, beans, and avocado), while the columns represent the interacting genes. The color gradient reflects the number of interactions, with darker shades of red indicating a higher frequency of interactions and blue representing no interaction. Abbreviations: *BAX*, BCL2 Associated X Protein; *BCL2*, B-cell Lymphoma 2; *BDNF*, Brain-Derived Neurotrophic Factor; *CASP3*, Caspase 3; *CAT*, Catalase; *GSR*, Glutathione-Disulfide Reductase; *HMOX1*, Heme Oxygenase 1; *IFNG*, Interferon Gamma; *IRS1*, Insulin Receptor Substrate 1; *JUN*, Jun Proto-Oncogene; *KCNJ2*, Potassium Inwardly Rectifying Channel Subfamily J Member 2; *KCNJ3*, Potassium Inwardly Rectifying Channel Subfamily J Member 3; *KCNJ6*, Potassium Inwardly Rectifying Channel Subfamily J Member 6; *MAPK1*, Mitogen-Activated Protein Kinase 1; *MAPK3*, Mitogen-Activated Protein Kinase 3; *MTR*, 5-Methyltetrahydrofolate-Homocysteine Methyltransferase; *NOS2*, Nitric Oxide Synthase 2; *NPY*, Neuropeptide Y; *PPARA*, Peroxisome Proliferator-Activated Receptor Alpha; *SLC11A2*, Solute Carrier Family 11 Member 2; *SLC1A2*, Solute Carrier Family 1 Member 2; *SOD1*, Superoxide Dismutase 1; *STK11*, Serine/Threonine Kinase 11; *TNF*, Tumor Necrosis Factor; *TP73*, Tumor Protein p73; *ACACA*, Acetyl-CoA Carboxylase Alpha; *AKT1*, AKT Serine/Threonine Kinase 1; *CCL2*, C-C Motif Chemokine Ligand 2; *FASN*, Fatty Acid Synthase; *IL1B*, Interleukin 1 Beta; *IL6*, Interleukin 6; *NFE2L2*, Nuclear Factor, Erythroid 2 Like 2 (NRF2); *NQO1*, NAD(P)H Quinone Dehydrogenase 1; *RELA*, RELA Proto-Oncogene, NF-κB Subunit; *TGFB1*, Transforming Growth Factor Beta 1; *FOS*, Fos Proto-Oncogene; *TP53*, Tumor Protein p53; *FADS2*, Fatty Acid Desaturase 2; *HMGCR*, 3-Hydroxy-3-Methylglutaryl-CoA Reductase; *HSD17B7*, Hydroxysteroid 17-Beta Dehydrogenase 7; *LPIN1*, Lipin 1; *PTEN*, Phosphatase and Tensin Homolog; *PTGS2*, Prostaglandin-Endoperoxide Synthase 2 (COX-2); *SIRT1*, Sirtuin 1; *AHR*, Aryl Hydrocarbon Receptor; *CCND1*, Cyclin D1; *CYP1B1*, Cytochrome P450 Family 1 Subfamily B Member 1; *ESR1*, Estrogen Receptor 1; *HMGB1*, High Mobility Group Box 1.

**Figure 4 ijms-26-08977-f004:**
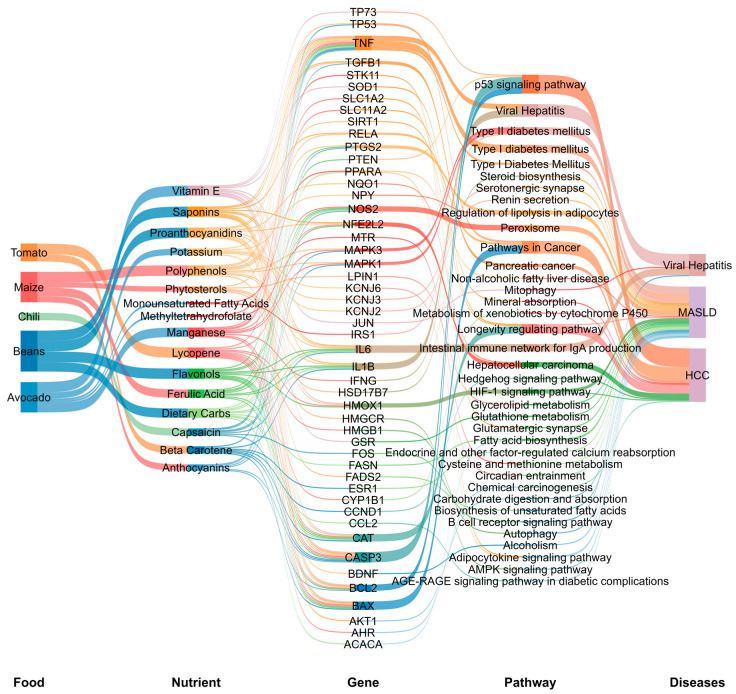
Sankey diagram linking Mexican food bioactive compounds, target genes, KEGG pathways, and liver diseases. The diagram displays the relationships between five culturally relevant foods (tomato, maize, chili, beans, and avocado) and their most relevant bioactive compounds (nutrient panel), the top 10 predicted target genes per nutrient (CTD), and enriched KEGG pathways (STRING, DAVID, and Enrichr) with liver diseases (MASLD, HCC, and viral hepatitis). Node categories are arranged from left to right: Food → Nutrient → Gene → Pathway → Disease. Flow thickness represents the number of connections between elements (greater thickness = more frequent associations). The color palette is applied solely to enhance interpretability and visual contrast between nodes, and does not convey biological meaning. Abbreviatures: CTD, Comparative Toxicogenomics Database; KEGG, Kyoto Encyclopedia of Genes and Genomes; STRING, Search Tool for the Retrieval of Interacting Genes/Proteins; DAVID, Database for Annotation, Visualization and Integrated Discovery; MASLD, Metabolic dysfunction-Associated Steatotic Liver Disease; HCC, Hepatocellular Carcinoma; *TP73*, Tumor Protein p73; *TP53*, Tumor Protein p53; *TNF*, Tumor Necrosis Factor; *TGFB1*, Transforming Growth Factor Beta 1; *STK11*, Serine/Threonine Kinase 11; *SOD1*, Superoxide Dismutase 1; *SLC1A2*, Solute Carrier Family 1 Member 2; *SLC11A2*, Solute Carrier Family 11 Member 2; *SIRT1*, Sirtuin 1; *RELA*, RELA Proto-Oncogene, NF-κB Subunit; *PTGS2*, Prostaglandin-Endoperoxide Synthase 2 (COX-2); *PTEN*, Phosphatase and Tensin Homolog; *PPARA*, Peroxisome Proliferator-Activated Receptor Alpha; *NQO1*, NAD(P)H Quinone Dehydrogenase 1; *NPY*, Neuropeptide Y; *NOS2*, Nitric Oxide Synthase 2; *NFE2L2*, Nuclear Factor, Erythroid 2 Like 2 (NRF2); *MTR*, 5-Methyltetrahydrofolate-Homocysteine Methyltransferase; *MAPK3*, Mitogen-Activated Protein Kinase 3; *MAPK1*, Mitogen-Activated Protein Kinase 1; *LPIN1*, Lipin 1; *KCNJ6*, Potassium Inwardly Rectifying Channel Subfamily J Member 6; *KCNJ3*, Potassium Inwardly Rectifying Channel Subfamily J Member 3; *KCNJ2*, Potassium Inwardly Rectifying Channel Subfamily J Member 2; *JUN*, Jun Proto-Oncogene; *IRS1*, Insulin Receptor Substrate 1; *IL6*, Interleukin 6; *IL1B*, Interleukin 1 Beta; *IFNG*, Interferon Gamma; *HSD17B7*, Hydroxysteroid 17-Beta Dehydrogenase 7; *HMOX1*, Heme Oxygenase 1; *HMGCR*, 3-Hydroxy-3-Methylglutaryl-CoA Reductase; *HMGB1*, High Mobility Group Box 1; *GSR*, Glutathione-Disulfide Reductase; *FOS*, Fos Proto-Oncogene; *FASN*, Fatty Acid Synthase; *FADS2*, Fatty Acid Desaturase 2; *ESR1*, Estrogen Receptor 1; *CYP1B1*, Cytochrome P450 Family 1 Subfamily B Member 1; *CCND1*, Cyclin D1; *CCL2*, C-C Motif Chemokine Ligand 2; *CAT*, Catalase; *CASP3*, Caspase 3; *BDNF*, Brain-Derived Neurotrophic Factor; *BCL2*, B-cell Lymphoma 2; *BAX*, BCL2 Associated X Protein; *AKT1*, AKT Serine/Threonine Kinase 1; *AHR*, Aryl Hydrocarbon Receptor; *ACACA*, Acetyl-CoA Carboxylase Alpha.

**Figure 5 ijms-26-08977-f005:**
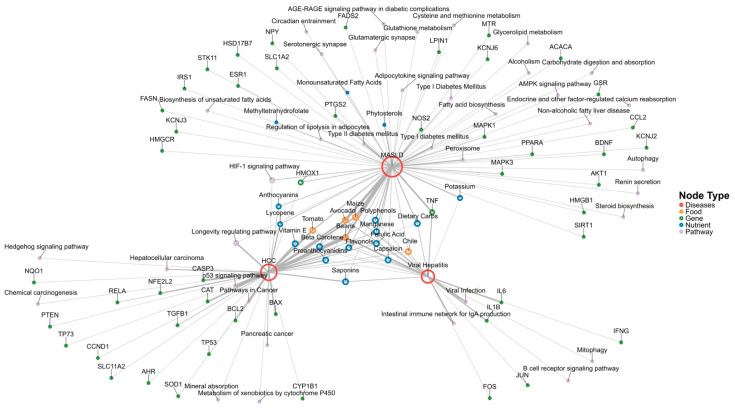
Network plot linking traditional Mexican foods, bioactive nutrients, target genes, enriched KEGG pathways, and liver diseases. The undirected network was generated using the Fruchterman–Reingold force-directed algorithm to cluster highly connected nodes visually. Nodes represent distinct biological or nutritional entities and are color-coded by category: red = diseases, orange = foods, blue = nutrients, green = genes, and purple = pathways. Node size is proportional to degree centrality (number of direct connections), highlighting hub elements within the network. Gray edges denote documented associations, with edge thickness proportional to co-occurrence frequency (weight) between each node and disease term. This representation emphasizes integrative links and multi-target relationships through which diet-derived bioactives may influence liver disease-related molecular pathways. Abbreviations: MASLD, Metabolic dysfunction-Associated Steatotic Liver Disease; TP73, Tumor Protein p73; TP53, Tumor Protein p53; TNF, Tumor Necrosis Factor; TGFB1, Transforming Growth Factor Beta 1; STK11, Serine/Threonine Kinase 11; SOD1, Superoxide Dismutase 1; SLC1A2, Solute Carrier Family 1 Member 2; SLC11A2, Solute Carrier Family 11 Member 2; SIRT1, Sirtuin 1; RELA, RELA Proto-Oncogene, NF-κB Subunit; PTGS2, Prostaglandin-Endoperoxide Synthase 2; PTEN, Phosphatase and Tensin Homolog; PPARA, Peroxisome Proliferator-Activated Receptor Alpha; NQO1, NAD(P)H Quinone Dehydrogenase 1; NPY, Neuropeptide Y; NOS2, Nitric Oxide Synthase 2; NFE2L2, Nuclear Factor Erythroid 2 Like 2; MTR, 5-Methyltetrahydrofolate-Homocysteine Methyltransferase; MAPK3, Mitogen-Activated Protein Kinase 3; MAPK1, Mitogen-Activated Protein Kinase 1; LPIN1, Lipin 1; KCNJ6, Potassium Inwardly Rectifying Channel Subfamily J Member 6; KCNJ3, Potassium Inwardly Rectifying Channel Subfamily J Member 3; KCNJ2, Potassium Inwardly Rectifying Channel Subfamily J Member 2; JUN, Jun Proto-Oncogene; IRS1, Insulin Receptor Substrate 1; IL6, Interleukin 6; IL1B, Interleukin 1 Beta; IFNG, Interferon Gamma; HSD17B7, Hydroxysteroid 17-Beta Dehydrogenase 7; HMOX1, Heme Oxygenase 1; HMGCR, 3-Hydroxy-3-Methylglutaryl-CoA Reductase; HMGB1, High Mobility Group Box 1; GSR, Glutathione-Disulfide Reductase; FOS, Fos Proto-Oncogene; FASN, Fatty Acid Synthase; FADS2, Fatty Acid Desaturase 2; ESR1, Estrogen Receptor 1; CYP1B1, Cytochrome P450 Family 1 Subfamily B Member 1; CCND1, Cyclin D1; CCL2, C-C Motif Chemokine Ligand 2; CAT, Catalase; CASP3, Caspase 3; BDNF, Brain-Derived Neurotrophic Factor; BCL2, B-Cell Lymphoma 2; BAX, BCL2 Associated X Protein; AKT1, AKT Serine/Threonine Kinase 1; AHR, Aryl Hydrocarbon Receptor; ACACA, Acetyl-CoA Carboxylase Alpha; IgA, Immunoglobulin A.

**Figure 6 ijms-26-08977-f006:**
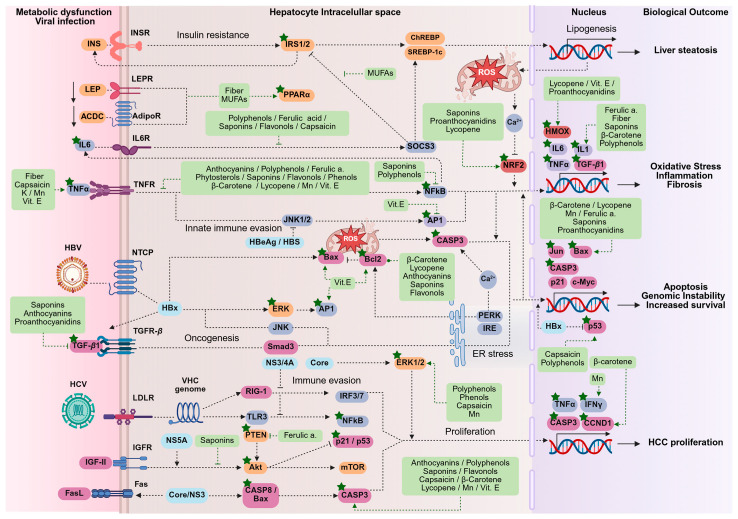
Nutrigenomic interactions of traditional Mexican foods in key hepatocyte pathways: potential implications for MASLD, viral hepatitis, and HCC. The diagram integrates viral infection mechanisms and metabolic dysfunction pathways within hepatocytes, highlighting molecular nodes targeted by bioactive compounds from Mexican food staples. Bioactive components such as polyphenols, dietary fiber, saponins, capsaicin, anthocyanins, lycopene, β-carotene, ferulic acid, phytosterols, flavonols, proanthocyanidins, manganese, monounsaturated fatty acids (MUFAs), and vitamin E are mapped to specific signaling cascades. Color coding indicates pathway categories: yellow = metabolic regulation, blue = immune modulation, red = oxidative stress response, and purple = proliferation/apoptosis. Green boxes denote bioactive compounds acting on each molecular target. Arrows indicate activation or inhibitory effects based on the literature and bioinformatic evidence, providing a mechanistic link between dietary patterns and modulation of liver disease pathways. Abreviations: INS, Insulin; INSR, Insulin Receptor; IRS1/2, Insulin Receptor Substrate 1/2; ChREBP, Carbohydrate Response Element-Binding Protein; SREBP-1c, Sterol Regulatory Element-Binding Protein 1c; ROS, Reactive Oxygen Species; LEP, Leptin; ACDC, Adiponectin; AdipoR, Adiponectin Receptor; LEPR, Leptin Receptor; PPARA, Peroxisome Proliferator-Activated Receptor Alpha; SOCS3, Suppressor of Cytokine Signaling 3; NRF2, Nuclear Factor Erythroid 2 Like 2; IL6, Interleukin 6; TNFα, Tumor Necrosis Factor Alpha; TNFR, Tumor Necrosis Factor Receptor; HMOX, Heme Oxygenase; IL1, Interleukin 1; TGF-β1, Transforming Growth Factor Beta 1; JUN, Jun Proto-Oncogene; BAX, BCL2 Associated X Protein; CASP3, Caspase 3; p21, Cyclin-Dependent Kinase Inhibitor 1A; c-Myc, Myc Proto-Oncogene; CCND1, Cyclin D1; IFNγ, Interferon Gamma; NFκB, Nuclear Factor Kappa B; AP1, Activator Protein 1; Vit. E, Vitamin E; PERK, Protein Kinase R-Like Endoplasmic Reticulum Kinase; IRE, Inositol-Requiring Enzyme 1; ER, Endoplasmic Reticulum; ERK, Extracellular Signal-Regulated Kinase; HBx, Hepatitis B Virus X Protein; NTCP, Sodium Taurocholate Cotransporting Polypeptide; HBV, Hepatitis B Virus; HCV, Hepatitis C Virus; VHC, Viral Hepatitis C; LDLr, Low-Density Lipoprotein Receptor; IGFR, Insulin-Like Growth Factor Receptor; Fas, Fas Receptor; FasL, Fas Ligand; IGF-II, Insulin-Like Growth Factor II; NS5A, Non-Structural Protein 5A; Core/NS3, Hepatitis C Virus Core and Non-Structural Protein 3; CASP8, Caspase 8; mTOR, Mechanistic Target of Rapamycin; p53, Tumor Protein p53; IRF3/7, Interferon Regulatory Factors 3 and 7; ERK1/2, Extracellular Signal-Regulated Kinases 1 and 2; TLR3, Toll-Like Receptor 3; PTEN, Phosphatase and Tensin Homolog; Ferulic a., Ferulic Acid; HCC, Hepatocellular Carcinoma; HBeAf, Hepatitis B e Antigen; HBs, Hepatitis B Surface Antigen; JNK1/2, c-Jun N-terminal Kinases 1 and 2; BCL2, B-cell Lymphoma 2.

**Figure 7 ijms-26-08977-f007:**
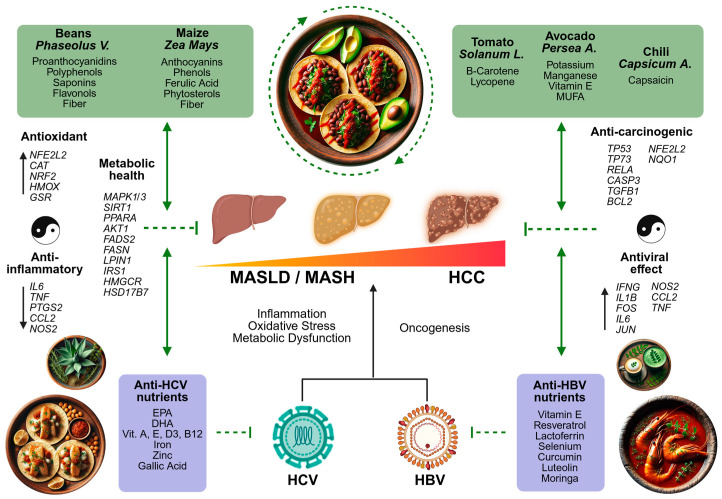
Nutrigenomic framework of traditional Mexican foods for modulating pathways involved in viral hepatitis and MASLD/MASH. The upper green panels present the traditional Mexican foods analyzed in the Functional Enrichment Analysis—beans (*Phaseolus V.*), maize (*Zea mays*), tomato (*Solanum* L.), avocado (*Persea americana*), and chili (*Capsicum annuum*)—along with their principal gene interactions identified through bioinformatic integration. These genes are implicated in antioxidant defense, metabolic regulation, anti-inflammatory activity, and anticancer pathways. The central section shows the key molecular targets linking these foods to protective effects against oxidative stress, inflammation, metabolic dysfunction, and carcinogenesis. The lower blue panels depict additional nutrients and bioactive compounds with activity against HCV and HBV. These antiviral compounds can be incorporated as complementary ingredients alongside the foods from the enrichment analysis to design culturally relevant dishes aimed at preventing liver injury and slowing its progression from both viral and metabolic etiologies. Abbreviatures: MUFA, Monounsaturated Fatty Acids; NFE2L2, Nuclear Factor Erythroid 2 Like 2 (NRF2); CAT, Catalase; HMOX1, Heme Oxygenase 1; GSR, Glutathione-Disulfide Reductase; MAPK1/3, Mitogen-Activated Protein Kinase 1 and 3 (ERK2/ERK1); SIRT1, Sirtuin 1; PPARA, Peroxisome Proliferator-Activated Receptor Alpha; AKT1, AKT Serine/Threonine Kinase 1; FADS2, Fatty Acid Desaturase 2; FASN, Fatty Acid Synthase; LPIN1, Lipin 1; IRS2, Insulin Receptor Substrate 2; HMGCR, 3-Hydroxy-3-Methylglutaryl-CoA Reductase; HSD17B7, Hydroxysteroid 17-Beta Dehydrogenase 7; IL6, Interleukin 6; TNF, Tumor Necrosis Factor; PTGS2, Prostaglandin-Endoperoxide Synthase 2 (COX-2); CCL2, C-C Motif Chemokine Ligand 2; NOS2, Nitric Oxide Synthase 2; HCV, Hepatitis C Virus; HBV, Hepatitis B Virus; HCC, Hepatocellular Carcinoma; MASLD, Metabolic dysfunction-Associated Steatotic Liver Disease; MASH, Metabolic dysfunction-Associated Steatohepatitis; EPA, Eicosapentaenoic Acid; DHA, Docosahexaenoic Acid; Vitamin A, Retinol; Vitamin E, α-Tocopherol; Vitamin D3, Cholecalciferol; Vitamin B12, Cobalamin; TP53, Tumor Protein p53; TP73, Tumor Protein p73; RELA, RELA Proto-Oncogene, NF-κB Subunit; CASP3, Caspase 3; TGFB1, Transforming Growth Factor Beta 1; BCL2, B-cell Lymphoma 2; NQO1, NAD(P)H Quinone Dehydrogenase 1; IFNG, Interferon Gamma; IL1B, Interleukin 1 Beta; FOS, Fos Proto-Oncogene; JUN, Jun Proto-Oncogene.

## Data Availability

The original contributions presented in this study are included in the article/Appendix A. Further inquiries can be directed to the corresponding author.

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
