# Peer review of "Genome-Based Mexican Diet Bioactives Target Molecular Pathways in HBV, HCV, and MASLD: A Bioinformatic Approach for Liver Disease Prevention"

_ijms, 2025, doi:10.3390/ijms26188977_

Round 1
Reviewer 1 Report
Comments and Suggestions for Authors
Dear Editor,
Thank you for the opportunity to review the manuscript entitled “Genome-Based Mexican Diet (GENOMEX) bioactives target molecular pathways in HBV, HCV, and MASLD: A bioinformatic approach for liver disease prevention”. This study presents a well-structured and thorough integrative analysis combining literature review and bioinformatics to elucidate the molecular mechanisms by which traditional Mexican diet bioactives may modulate viral hepatitis and metabolic liver disease pathways. The manuscript is relevant and timely, addressing an important public health issue with a culturally tailored nutritional approach.
The paper has a clear and logical presentation of objectives and methodology, with an effective integration of genomic databases with nutritional data, and a comprehensive identification of gene-nutrient interactions relevant to liver disease with a discussion linked to clinical and public health implications.
In any case, there are some limitations and suggestions: 1) a greater clarity in some bioinformatic methods would improve reproducibility 2) the discussion could further address limitations of in silico findings and need for experimental validation 3) you can consider expanding on potential population-specific factors and limitations for wider applicability.
In conclusion, I recommend acceptance pending minor revisions to enhance methodological transparency and contextual discussion.
Author Response
Thank you very much for taking the time to review this manuscript. Please find the detailed point-by-point responses below, and the corresponding revisions/corrections highlighted in track changes and marked in yellow in the re-submitted files.
Comment 1:
A greater clarity in some bioinformatic methods would improve reproducibility.
Response 1:
To improve the transparency and reproducibility of our bioinformatic workflow, we have revised the “Integrative Bioinformatic Analysis” section of the Methods (page 4-5, line 170-194). We now provide additional details on input data selection, analysis parameters, thresholds, and visualization steps, including specific reference to how gene-nutrient interactions were filtered, enrichment significance was determined (e.g., FDR threshold), and which R packages, libraries and functions were used
- 2.2 Identification of gene interactions – Added details on CTD query, selection of top 10 genes per nutrient/bioactive compound, and criteria for biological relevance (Page 5, lines 287–297).
- 2.3 Functional Enrichment Analysis – Specified enrichment tools (STRING, DAVID, Enrichr), statistical tests, FDR correction, and significance thresholds (Page 5, lines 299–310).
- 2.4 Data Visualization – Described three visualization approaches (heatmap, Sankey diagram, weighted network), R packages used, color coding, and meaning of graphical elements (Pages 5–6, lines 312–333).
Comment 2:
The discussion could further address limitations of in silico findings and need for experimental validation.
Response 2:
We appreciate the reviewer’s suggestion. In response, we have expanded the discussion to more clearly acknowledge the limitations of in silico approaches and the necessity of experimental validation. (Page 20-21, Line 1050-1054).
Comment 3:
You can consider expanding on potential population-specific factors and limitations for wider applicability.
Response 3:
We thank the reviewer for this suggestion. To address this point, we have expanded the discussion to explicitly include potential population-specific limitations that may affect the wider applicability of our findings. A new paragraph was added in the Discussion emphasizing that nutrient–gene interactions are influenced by population-level differences in ancestry, genetic polymorphisms, epigenetic patterns, microbiome composition, and cultural dietary practices. We also acknowledge that while this study focuses on the Mexican population, additional research is needed to validate whether the observed interactions are generalizable to other populations with distinct genetic backgrounds and nutritional environments (page 21, lines 1061-1072).
Reviewer 2 Report
Comments and Suggestions for Authors
Dear authors, While the topic of your study is very interesting and your methodology elaborately described, I appreciate the confounders from environmental exposure. Otherwise, I have highlighted some minor points to address.

Author Response
Dear authors, While the topic of your study is very interesting and your methodology elaborately described, I appreciate the confounders from environmental exposure. Otherwise, I have highlighted some minor points to address.
Response to reviewer 2
Thank you very much for taking the time to review this manuscript. Please find the detailed point-by-point responses below, and the corresponding revisions/corrections highlighted in track changes and marked in yellow in the re-submitted files.
Comment 1:
1.- Abstract lacks clarity in the results and discussion section.
Response:
We appreciate the reviewer’s observation. To improve clarity, we have revised the Results section of the abstract to clearly summarize the main findings of our integrative analysis. (Page 1, lines 29–35)
Comment 2:
Metabolic related diseases: Metabolic syndrome or its components and related comorbidities, rather than diseases-metabolic liver diseases includes haemochromatosis, wilsons etc apart from MS.
Response 2:
Thank you for pointing out this distinction. We agree that the term “metabolic-related diseases” could be misleading, as “metabolic liver diseases” often encompasses hereditary conditions such as haemochromatosis or Wilson’s disease. To avoid this confusion, we have rephrased the sentence to clearly refer to metabolic syndrome and its associated comorbidities, which aligns with the focus of our study. (page 2, line 82-87)
Comment 3:
Rephrase line 56 (advanced countries).
Response 3:
We agree that the term “advanced countries” is imprecise and may be outdated. To improve clarity and align with current epidemiological terminology, we have revised the phrase to high-income countries (page 2, line 88).
Comment 4:
clarify epidemiological transition
Response 4:
We agree that the term “epidemiological transition” required contextual clarification. We have revised the sentence to specify that the transition refers to the Mexican population, where a shift from communicable to non-communicable diseases is underway (page 2, line 91-94).
Comment 5:
-chance singly in line 68.
Response 5:
We agree that the term “singly” was unclear in this context. To improve clarity and readability, we have rephrased the sentence using more conventional language. (page 2, lines 102-103)
Comment 6:
-clarify the escalating intravenous drug use in Mexico.
Response 6:
Thank you for this important point. We agree that the role of intravenous drug use as a risk factor for hepatitis B and C infection in Mexico required clearer explanation. We have rephrased the sentence to explicitly link this behavior to viral hepatitis transmission (page 2, line 104-105).
Comment 7:
- Not all BMI >25 are at risk for MASH, do you have an actual risk number for Mexico? Preferably for MASLD and MASH separately.
Response 7:
In response, we revised the manuscript to clarify that not all individuals with BMI >25 kg/m² are at risk for MASLD or MASH. To strengthen this point, we added national prevalence estimates for MASLD in Mexico and included findings from our research group that highlight the presence of liver disease even among normal weight individuals. This underscores that metabolic risk and liver damage are not exclusively associated with being overweight or obesity. These additions help contextualize the heterogeneity of MASLD risk in the Mexican population. The updated text appears in the revised manuscript on page 2, lines 105–115.
Comment 8:
-elaborate more the GENOMEX intervention in line 81.
Response 8:
We appreciate this comment and agree that readers unfamiliar with the GENOMEX concept would benefit from a clearer explanation. We have revised the introduction to briefly define the GENOMEX intervention as a genome-based, culturally tailored nutritional strategy developed in Mexico. The updated text highlights its integration of traditional food sources with genomic and clinical profiling, as well as prior evidence of its effectiveness in improving metabolic risk factors and modulating antiviral pathways. (Page 3, Lines 142-152).
Comment 9:
Line 88 – change ‘enables’ to ‘single’.
Response 9:
We agree with your observation and have revised the sentence to improve both clarity and grammar. Therefore, the sentence now reads: Furthermore, bioinformatics enables the comprehensive integration of molecular interactions across the genomic landscape, revealing network characteristics intricately linked to disease pathogenesis. This correction appears in page 3, line 156 of the revised manuscript.
Comment 9:
Line 211: Do you have a table saying why you rejected the studies that you decided not to include? Where have you stated the established criteria?
Response 9:
We have now clarified both the inclusion and exclusion criteria within the Methods section. While we did not generate a formal table listing all excluded studies, we have now detailed the Boolean search strategy and reported the number of articles identified through PubMed and Google Scholar searches, as well as how many remained after applying the criteria (Pages 3-4, lines 187-212).
Comment 10:
In which setting? Please clarify accordingly for all your citations. It’s different if this is a clinical correlation (confounders) vs in vitro study (much clearer causative relationship but not sure if clinically relevant).
Response 10:
We agree that the experimental context, whether in vitro, in vivo, or clinical—has a critical impact on how findings are interpreted and their potential translational relevance. Although this information was detailed in Supplementary Table S1, it was not sufficiently highlighted in the main text. To address this, we have revised Sections 3.1.1 Antiviral Nutrients and 3.1.2 Anti-MASLD Effects (Pages 7–11, Lines 441–639) to explicitly state the study setting for each cited finding, thereby enhancing clarity and helping the reader better assess the strength and clinical
applicability of the evidence.
Comment 11:
Line 263: Anti-MASLD effects—Expand on the role of the Mexican diet on intestinal flora and its implications in MASLD (very well proven link). All the fiber and healthy fatty acids you describe must influence gut microbiota and have a positive impact on fatty liver disease.
Response 11:
To further strengthen our manuscript, we have expanded on the role of dietary fibers (prebiotics) and healthy fatty acids from traditional Mexican foods in modulating gut microbiota and their beneficial implications for MASLD. We specifically highlight recent evidence demonstrating how prebiotics promote beneficial bacterial growth, increase short-chain fatty acid (SCFA) production, improve gut barrier function, and subsequently reduce hepatic inflammation, steatosis, and fibrosis. (page 11, lines 641-667).
Reviewer 3 Report
Comments and Suggestions for Authors
This manuscript is well-written and addresses an important topic at the intersection of viral hepatitis, MASLD, and hepatocellular carcinoma. The combination of literature review and bioinformatic analysis provides valuable insights into potential nutrigenomic interventions.
Major
- Most findings are derived from in silico, in vitro, or animal studies. The conclusions should be tempered to acknowledge the limited number of clinical trials and avoid overstating translational readiness.
- Several bioactives (e.g., resveratrol, curcumin, EGCG) are typically studied in supplement form rather than from whole-food consumption. The manuscript should clarify this distinction.
- Figures 4–7 are visually complex. Simplifying legends and adding a more schematic summary would aid reader comprehension. In addition, the figure legends should provide more detailed explanations of the pathways, gene interactions, and interpretation of color scales and flow thickness so that readers can fully understand the meaning without referring back to the text.
- Expand this section to highlight variability in study designs, lack of dietary intervention trials in Mexican populations, and socioeconomic or agricultural barriers (e.g., loss of native maize varieties).
- The manuscript would benefit from a discussion on how the GENOMEX diet compares to dietary patterns in other countries (e.g., Mediterranean, Asian, or North American diets). It should be clarified whether the Mexican diet provides unique or superior bioactive profiles, or whether these effects are shared across global traditional diets. This context will help readers understand the distinctiveness and broader relevance of the GENOMEX approach.
Author Response
Thank you very much for taking the time to review this manuscript. Please find the detailed point-by-point responses below, and the corresponding revisions/corrections highlighted in track changes and marked in yellow in the re-submitted files.
Comment 1:
Most findings are derived from in silico, in vitro, or animal studies. The conclusions should be tempered to acknowledge the limited number of clinical trials and avoid overstating translational readiness.
Response 1:
We appreciate this observation. In response, we have clarified in the Results (Pages 7–11, Lines 441–639) and Discussion sections that a part of the evidence presented in this study stems from in silico, in vitro, or animal research, and that these findings should be interpreted as preliminary. We have expanded the limitations paragraph (Pages 20-21, Lines 1050-1054) to explicitly state that while bioinformatic tools are valuable for hypothesis generation, however, they do not replace the need for experimental validation. We also acknowledge the current lack of large-scale clinical trials evaluating the health effects of traditional Mexican foods within a nutrigenomic framework (Pages 22, Lines 1021). These additions temper the conclusions and reinforce the need for future validation in human populations. Additionally, we have revised the Conclusion to emphasize the potential of integrating traditional Mexican foods into nutrigenomic strategies, rather than implying immediate clinical translation (Pages 22, Lines 1121).
Comment 2:
Several bioactives (e.g., resveratrol, curcumin, EGCG) are typically studied in supplement form rather than from whole-food consumption. The manuscript should clarify this distinction.
Response 2:
We have revised the Discussion section to clarify that many of the bioactive compounds analyzed in this study, such as resveratrol, curcumin, and EGCG, have primarily been studied in supplement or purified forms. We now note that their absorption, metabolism, and physiological effects may vary significantly when consumed as part of whole foods, due to the influence of the food matrix and synergistic nutrient interactions (Page 20, Lines 1056-1058).
Comment 3:
Figures 4–7 are visually complex. Simplifying legends and adding a more schematic summary would aid reader comprehension. In addition, the figure legends should provide more detailed explanations of the pathways, gene interactions, and interpretation of color scales and flow thickness so that readers can fully understand the meaning without referring back to the text.
Response 3:
We thank the reviewer for this helpful suggestion. In response, we have revised the legends of Figures 4–7 to improve clarity, reduce redundancy, and provide self-contained descriptions that allow readers to understand each figure without referring to the main text. For each figure, we have included:
- A concise schematic summary of what the figure represents.
- A clear explanation of node/element categories and how they are arranged.
- The meaning of colors.
- The interpretation of flow thickness or edge width in relation to interaction frequency or connectivity.
- Key biological interpretation linking visualization to the study’s objectives.
The updated figure legends are as follows:
- Figure 4 (Page 13, Lines 719-727): Legend revised to simplify description, clarify node arrangement, explain meaning of flow thickness, and state that colors are for visual contrast only.
- Figure 5 (Page 14, Lines 761-770): Legend updated to include schematic summary, node category colors, meaning of node size and edge thickness, and clarification that colors have no biological meaning.
- Figure 6 (Page 15, Lines 804-814): Legend expanded to detail pathway categories, bioactive compound mapping, and meaning of symbols (green boxes, arrows).
- Figure 7 (Page 18, Lines 909-920): Legend rewritten to clarify panel content, gene interactions, and role of additional antiviral compounds; notes potential use for prevention and as adjunct treatment.
Comment 4:
Expand this section to highlight variability in study designs, lack of dietary intervention trials in Mexican populations, and socioeconomic or agricultural barriers (e.g., loss of native maize varieties).
Response 4:
We thank the reviewer for this recommendation. In response, we have expanded the limitations section of the Discussion to address several critical contextual factors. We now explicitly acknowledge the variability in study designs across the literature and the lack of dietary intervention trials specifically conducted in Mexican populations, which limits the generalizability and direct translational application of our findings. Additionally, we have incorporated a discussion of socioeconomic and agricultural barriers—such as unequal access to traditional foods, shifts in dietary patterns, and the progressive loss of native maize varieties—which may constrain the implementation of GENOMEX-based strategies. These additions provide a more realistic framework for interpreting the results and reinforce the need for context-specific, population-based validation (Page 21, Lines 1059-1072).
Comment 5:
The manuscript would benefit from a discussion on how the GENOMEX diet compares to dietary patterns in other countries (e.g., Mediterranean, Asian, or North American diets). It should be clarified whether the Mexican diet provides unique or superior bioactive profiles, or whether these effects are shared across global traditional diets. This context will help readers understand the distinctiveness and broader relevance of the GENOMEX approach.
Response 5:
We appreciate this suggestion. In response, we have expanded the Discussion section to contextualize the GENOMEX approach in relation to other global dietary patterns. Specifically, we now clarify that while the GENOMEX diet shares functional properties with other traditional diets, it is not positioned as universally superior. Rather, our focus is to advocate for a culturally and genetically tailored nutritional framework that addresses the specific needs of the Mexican population (Pages 21, Lines 1087-1096).
Reviewer 4 Report
Comments and Suggestions for Authors
I read with interest the manuscript entitled "Genome-Based Mexican Diet (GENOMEX) bioactives target molecular pathways in HBV, HCV, and MASLD: A bioinformatic approach for liver disease prevention"
It is not acceptable to use abbreviations in the title. Please correct it.
Given the set goal of the study, it is necessary to conduct a systematic literature review using at least 3 databases such as PubMed, Scopus and WOS. The study in your form is of low quality and low level of evidence.
Please register the systematic review in the PROSPERO database. Please follow the detailed conditions for conducting a systematic review because your methodology is quite insufficient.
Figure 1 is insufficient. The PRISMA flowchart needs to be formatted.
It is unclear how many studies were included, based on what criteria, how many researchers conducted the analysis, etc.
Although I do not doubt your results, they must come from a methodically scientifically based study.
It is not customary for figures to be part of the discussion.
While I appreciate your effort in this study, in view of the high quality of the journal, I ask that you conduct a systematic review on the topic in question following the guidelines in writing.
Author Response
I read with interest the manuscript entitled "Genome-Based Mexican Diet (GENOMEX) bioactives target molecular pathways in HBV, HCV, and MASLD: A bioinformatic approach for liver disease prevention"
1.- It is not acceptable to use abbreviations in the title. Please correct it.
2.- Given the set goal of the study, it is necessary to conduct a systematic literature review using at least 3 databases such as PubMed, Scopus and WOS. The study in your form is of low quality and low level of evidence.
3.- Please register the systematic review in the PROSPERO database. Please follow the detailed conditions for conducting a systematic review because your methodology is quite insufficient.
Figure 1 is insufficient. The PRISMA flowchart needs to be formatted.
4.- It is unclear how many studies were included, based on what criteria, how many researchers conducted the analysis, etc.
Although I do not doubt your results, they must come from a methodically scientifically based study.
5.- It is not customary for figures to be part of the discussion.
6.- While I appreciate your effort in this study, in view of the high quality of the journal, I ask that you conduct a systematic review on the topic in question following the guidelines in writing.
Response to Reviewer 4:
We sincerely thank Reviewer 4 for the time and effort dedicated to evaluating our manuscript. We appreciate your thorough review and valuable suggestions. Below, we provide a point-by-point response to your comments. We respectfully clarify that our manuscript does not aim to be a systematic review, but rather an integrative bioinformatic and nutrigenomic analysis based on curated literature and databases. Nonetheless we have improved transparency in the methodology and added relevant clarifications, as detailed below.
Comment 1:
It is not acceptable to use abbreviations in the title. Please correct it.
Response 1:
We have revised the title to remove the abbreviation "GENOMEX," as it may not be widely recognized by all readers. However, we have retained commonly accepted and official abbreviations such as MASLD, HCV, and HBV, as they are standard terminology in the field and improve title clarity (Page 1, line 2).
Comment 2:
Given the set goal of the study, it is necessary to conduct a systematic literature review using at least 3 databases such as PubMed, Scopus and WOS. The study in your form is of low quality and low level of evidence.
Response 2:
We respectfully clarify that our study is not intended to be a systematic review. Instead, we present an integrative nutrigenomic analysis combining curated literature with bioinformatic tools to explore molecular mechanisms linking traditional Mexican diet bioactives to liver disease related pathways. Our goal is to identify and visualize nutrient-gene-disease interactions to inform future experimental and clinical research in Mexico. Nevertheless, we have now clearly stated the nature and limitations of our literature search and analysis in the Limitations section (page 20, lines 1041-1054).
Comment 3:
Please register the systematic review in the PROSPERO database. Please follow the detailed conditions for conducting a systematic review because your methodology is quite insufficient. Figure 1 is insufficient. The PRISMA flowchart needs to be formatted.
Response 3:
As explained above, our manuscript does not follow a systematic review protocol and therefore does not qualify for PROSPERO registration. We have, however, improved transparency in the Methods section (page 3-4, lines 163-212) by specifying the databases used, general selection criteria, and limitations of our approach (page 20,21, lines 1041-1054).
Comment 4:
It is unclear how many studies were included, based on what criteria, how many researchers conducted the analysis, etc. Although I do not doubt your results, they must come from a methodically scientifically based study.
Response 4:
We appreciate the reviewer’s concern regarding methodological transparency. In response, we have expanded the Methods section (page 3, lines 166–177) to clearly state the inclusion and exclusion criteria used in the literature review, as well as the number of studies retrieved and retained. Specifically, we conducted structured Boolean searches in PubMed and Google Scholar using viral hepatitis, MASLD, and Mexican diet related terms, and applied predefined criteria to include only peer-reviewed original studies providing experimental, mechanistic, or clinical evidence related to traditional Mexican foods and HBV, HCV, or MASLD (Pages 3-4, lines 187-212). Additionally, we now specify that the review and screening process was conducted by two independent researchers with expertise in nutrigenomics and liver disease. Discrepancies were resolved by consensus (Page 3, lines 175-177).
Comment 5:
It is not customary for figures to be part of the discussion.
Response 5:
We acknowledge that including figures within the Discussion section is not standard practice. However, Figure 7 is central to integrating and interpreting our key findings—it visually synthesizes the nutrigenomic mechanisms identified and highlights their potential translational and clinical implications. For this reason, and to maintain its relevance and immediate accessibility to readers, we have decided to retain it in the Discussion section. To enhance clarity and appropriateness, we have substantially revised its caption to provide a more precise explanation of its conceptual role in interpreting the results (Page 18, lines 909-920).
Comment 6:
While I appreciate your effort in this study, in view of the high quality of the journal, I ask that you conduct a systematic review on the topic in question following the guidelines in writing.
Response 6:
We fully agree that a systematic review on this topic would be highly valuable. However, such an undertaking falls outside the scope and intent of the present study. Our manuscript was conceived as a conceptual and bioinformatic framework to identify priority molecular targets and diet-derived bioactives that warrant future investigation. We hope the revisions we have made clarify this distinction.
It is important to note that we have previously evaluated the culturally rooted Mexican dietary pattern in clinical settings, obtaining significant and positive results. Building on that evidence, the present study provides an accessible means to explore the underlying nutrigenomic mechanisms, helping to bridge clinical observations with molecular pathways.
Looking ahead, we will consider conducting systematic reviews and meta-analyses within this research line to further increase the reliability and robustness of future findings. Nevertheless, we believe that the current approach is both innovative and integrative, offering a strong foundation for promoting culturally and genomically tailored nutrition. By leveraging accessible bioinformatic tools, our work enables the exploration of nutrigenomic mechanisms—in this case, those of the traditional Mexican diet—thereby supporting the development of region-specific dietary strategies grounded in both genomic science and cultural heritage.
Round 2
Reviewer 2 Report
Comments and Suggestions for Authors
as discussed
Author Response
Dear Reviewer,
We appreciate your time and expertise to review this work.
Reviewer 3 Report
Comments and Suggestions for Authors
The authors have successfully addressed my concerns and questions.
Reviewer 4 Report
Comments and Suggestions for Authors
Thank you for the revised manuscript, but you must follow the transparency, rigor, and adherence to established reporting standards, especially given the high quality of the journal. Since you have elements of a systematic review in your manuscript, registration increases transparency and reduces bias. You must follow the PRISMA guidelines. This increases the credibility of the manuscript and aligns it with the journal's standards. Please follow the recommendation and resubmit the manuscript.
Author Response
Dear Reviewer,
We appreciate the feedback that all reviewers have provided to ensure clarity and reproducibility, which we have improved based on the comments in the revised version of the manuscript.
Specifically, for this last comment, we again would like to clarify that this work is a narrative review with an integrative bioinformatic analysis, designed to map mechanistic links between bioactives in the Mexican dietary pattern and molecular pathways implicated in HBV, HCV, and MASLD. Accordingly, we synthesize heterogeneous evidence (in silico, in vitro, in vivo, and clinical) and apply a bioinformatic enrichment workflow to examine the complex interactions of food matrices, connecting nutrients/bioactives, genes/proteins, and pathways, as detailed in the Methods and Results.
As noted in the Limitations, clinical evidence specifically evaluating traditional Mexican foods within medical-nutrition care remains scarce. Our review was conceived to elucidate nutrigenomic mechanisms by which nutrients and bioactives from these foods, successfully used in the GENOMEX clinical intervention, operate at the genomic level. Rather than ranking foods, we examine how multi-component foods engage liver-disease–related pathways, situating these mechanisms within a culturally grounded dietary pattern consumed for millennia by Native Mexican populations.
Nonetheless, to address the request for standards and transparency, we conducted an internal PRISMA crosswalk and, consistent with our study design, the manuscript already meets the transparency-oriented items appropriate to a narrative integrative review. Specifically, it articulates a clear rationale and objectives; prespecifies eligibility criteria tailored to the research question; names the information sources and reproduces the Boolean search strings for both antiviral and anti-MASLD queries, including per-database counts; describes a dual, independent screening process with consensus and without automation; and presents structured tabulation/figures that connect foods and bioactives to genes, pathways, and disease mechanisms. These elements ensure traceability of the evidence base and reproducibility of the literature component at the level warranted by this review.
By design, the manuscript follows the PRISMA elements relevant to a narrative review with integrative bioinformatics that guarantee a clear rationale, eligibility criteria, reproducible search strategies, independent screening, and structured figures/tables, while omitting Systematic Review (SR)-exclusive components that would misstate its aims. SR–exclusive components such as protocol registration, PRISMA-Abstract formatting, formal data-extraction workflows, risk-of-bias tools, predefined effect measures, quantitative pooling, heterogeneity or sensitivity analyses, reporting-bias assessments, or certainty grading (e.g., GRADE). These methods are indispensable when estimating pooled comparative effects across commensurable outcomes. However, they are neither appropriate for a mechanistic, multi-component synthesis whose purpose is to map biological pathways across heterogeneous in silico, in vitro, in vivo, and clinical evidence. Accordingly, this manuscript is not a systematic review and does not claim to be one. We respectfully ask that it be evaluated on its stated design and purpose: a transparent narrative review integrated with bioinformatic analysis.
Please note that we have made some edits in the corresponding section of limitations and strengths based on these comments (Page 20, lines 912-1025)